# Universal Approximation of Mean-Field Models via Transformers

**Shiba Biswal** [* 1]  **Karthik Elamvazhuthi** [* 1]  **Rishi Sonthalia** [* 2]

## Abstract

This paper investigates the use of transformers to approximate the mean-field dynamics of interacting particle systems exhibiting collective behavior. Such systems are fundamental in modeling phenomena across physics, biology, and engineering, including opinion formation, biological networks, and swarm robotics. The key characteristic of these systems is that the particles are indistinguishable, leading to permutation-equivariant dynamics. First, we empirically demonstrate that transformers are well-suited for approximating a variety of mean field models, including the Cucker-Smale model for flocking and milling, and the mean-field system for training two-layer neural networks. We validate our numerical experiments via mathematical theory. Specifically, we prove that if a finite-dimensional transformer effectively approximates the finite-dimensional vector field governing the particle system, then the $L_2$ distance between the *expected transformer* and the infinite-dimensional mean-field vector field can be uniformly bounded by a function of the number of particles observed during training. Leveraging this result, we establish theoretical bounds on the distance between the true mean-field dynamics and those obtained using the transformer.

## 1  Introduction

The identification of dynamical system models for physical processes is a fundamental application of machine learning (ML). Of particular interest are systems of particles exhibiting collective behaviors—such as swarming, flocking, opinion dynamics, and consensus. These systems involve a large number of particles or agents that follow identical dynamics, which are independent of the particles' identities and are *permutation-equivariant*. Examples include biological entities (Lopez et al., 2012), robots (Elamvazhuthi & Berman, 2019), traffic flow (Piccoli et al., 2009; Siri et al., 2021), and parameters in two-layer neural networks (Mei et al., 2019). A common approach to simplifying the analysis of such systems is to consider the continuum limit as the number of particles $n \to \infty$, resulting in *mean-field models* rooted in statistical physics. Instead of specifying the dynamics of each agent, the particles are modeled using probability measures. This paper learns the mean-field dynamics of particles via particle trajectories using transformers.

Let $\Omega \subset \mathbb{R}^d$. Consider a vector field $\mathcal{F} : \Omega \times \mathcal{P}(\Omega) \to \mathbb{R}^d$. The following equation describes the general mean-field behavior of interacting particles evolving on $\Omega$,

$$\frac{dz}{dt} = \mathcal{F}(z, \mu), \; z(0) = z_0 \sim \mu_0,$$

where $z \in \Omega$ denotes the state of a particle, $\mu \in \mathcal{P}(\Omega)$ is the distribution of the particles at time $t$, and $\mu_0$ is the initial distribution. The above equation can be approximated by a finite particle-level model. Let $\mathbf{z} = (z_1, \ldots, z_n) \in \Omega^n$ represent the state of $n$ particles, and consider the following,

$$\dot{z}_i = \mathcal{F}(z_i, \nu_{\mathbf{z}}^n),$$

where the empirical distribution of the $n$-particle system given by $\nu_{\mathbf{z}}^n := \frac{1}{n} \sum_{i=1}^{n} \delta_{z_i} \in \mathcal{D}^n(\Omega)$. Several works learn the mean-field dynamics by learning the particle-level dynamics. Pham & Warin (2023) constructed a neural network via binning as approximations of the vector field and proved approximation capabilities when restricted to dynamics on measures that admit densities. The works of Feng et al. (2024); Lu et al. (2019); Miller et al. (2023) present a kernel-based method for identifying the dynamics of interacting particle systems. Whereas, Messenger & Bortz (2022) presents a form of the SINDy algorithm for identifying mean-field dynamics of interacting particle systems. Kratsios et al. (2022) constructed a probabilistic transformer that can satisfy additional constraints via maps on measures. Additionally, Furuya et al. (2025) use a continuous version of transformers and attention (Vuckovic et al., 2020; Geshkovski et al., 2023) and provide universal approximation of measure theoretic maps. Similarly (Adu & Gharesifard, 2024) have proven an approximation result for

---

[*]Equal contribution [1]Theoretical Division, Los Alamos National Laboratory, Los Alamos, NM, USA [2]Boston College, Boston, MA, USA. Correspondence to: Shiba Biswal <sbiswal@lanl.gov>, Rishi Sonthalia <sonthal@bc.edu>.

*Proceedings of the 42nd International Conference on Machine Learning*, Vancouver, Canada. PMLR 267, 2025. Copyright 2025 by the author(s).

measure theoeretic maps arising from solutions of continuity equations on the sphere using a different architecture.

This paper explores the use of transformers to approximate the dynamical systems that govern the collective behavior of interacting particles with permutation-equivariant dynamics. We compare the transformer against models from Pham & Warin (2023) and Messenger & Bortz (2022) as well as two additional permutation equivariant baselines models on synthetic and real data from the Cucker-Smale model for swarming and milling. We prove theoretical guarantees for the transformer by lifting it from a sequence-to-sequence map to a map on measures, by taking the expectation of a finite-dimensional transformer with respect to a product measure. We refer to this as the *expected transformer*.

The main **contributions** are as follows:

1. We empirically show that transformers approximate the vector field better compared to other network architectures (see Table 1).
2. We define a continuum version of the transformer as an expectation of finite-dimensional transformers (see (9)).
3. We establish approximation rate bounds of measure-valued maps by this expected transformer (see Theorem 4.7).
4. We mathematically show that the solution to the mean-field model can be approximated by approximating the vector field by the expected transformer (see Theorem 4.14 and Figure 2).

The rest of the paper is organized as follows. Section 1.1 defines the notation used in the paper. Section 2 defines the problem, Section 3 presents our numerical experiments, and Section 4 presents our theoretical results.

### 1.1 Notation

Let $\mathbb{R}^d$ denote the $d$-dimensional Euclidean space, and let $\mathbb{Z}_+$ denote the set of positive integers. The diameter of a subset $A \subset \mathbb{R}^d$ is defined as $\operatorname{diam}(A) := \sup\{\|x - y\| : x, y \in A\}$ and let $B_r(z)$ be the closed ball of radius $r > 0$ centered at $z \in \mathbb{R}^d$.

We denote by $\mathcal{P}(\mathbb{R}^d)$ the set of all Borel probability measures on $\mathbb{R}^d$. Similarly, for a subset $\Omega \subset \mathbb{R}^d$, $\mathcal{P}(\Omega)$ denotes the set of Borel probability measures on $\Omega$. The subset of probability measures $\nu$ with finite $p$-th moments, $M_p(\nu) := \left(\int_{\mathbb{R}^d} \|x\|^p \, d\nu(x)\right)^{1/p}$ is denoted by $\mathcal{P}_p(\mathbb{R}^d) := \{\nu \in \mathcal{P}(\mathbb{R}^d) : M_p(\nu) < \infty\}$. Let $\mathcal{P}_c(\mathbb{R}^d)$ be the set of probability measures with compact support. The set of empirical measures formed by finite sums of $n$ Dirac-delta measures is denoted by $\mathcal{D}^n(\mathbb{R}^d) := \{\nu \in \mathcal{P}(\mathbb{R}^d) : \nu = \frac{1}{n}\sum_{i=1}^n \delta_{x_i}, \ x_i \in \mathbb{R}^d\}$. Similarly, the set $\mathcal{D}^n(\Omega)$ denotes the set of empirical measures on $\Omega \subset \mathbb{R}^d$. For $\nu \in \mathcal{P}(\mathbb{R}^d)$, the $n$-fold product measure on $(\mathbb{R}^d)^n$ is de-

noted by $\nu^{\otimes n} := \underbrace{\nu \times \cdots \times \nu}_{n \text{ times}}$. The support of a measure $\nu \in \mathcal{P}(\mathbb{R}^d)$, denoted by $\operatorname{supp}(\nu)$, is the smallest closed set $S \subset \mathbb{R}^d$ such that $\mu(\mathbb{R}^d \setminus S) = 0$. Given a measurable map $X : \mathbb{R}^d \to \mathbb{R}^d$ and a measure $\mu \in \mathcal{P}(\mathbb{R}^d)$, the *pushforward measure* $X_\#\mu \in \mathcal{P}(\mathbb{R}^d)$ is defined by $X_\#\mu(A) := \mu\left(X^{-1}(A)\right)$ for every Borel measurable set $A \subset \mathbb{R}^d$.

For a vector $y \in \mathbb{R}^d$, the $i$-th component is denoted by $y_i$. The $p$-norm, $p \in [1, \infty)$, and $\infty$-norm of $y$ are, respectively, $\|y\|_p^p = \sum_{i=1}^d |y_i|^p$, $\|y\|_\infty = \max_i |y_i|$. Boldface letters, such as $\mathbf{z}$, denote elements in $\mathbb{R}^{d \times n}$, representing collection of $n$ vectors in $\mathbb{R}^d$. The $i, j$-th element is denoted by $\mathbf{z}_{i,j}$. The corresponding $p$-norm ($p \in [1, \infty)$) and $\infty$-norm of $\mathbf{z}$ are, respectively, $\|\mathbf{z}\|_p^p = \sum_{i=1}^d \sum_{j=1}^n |\mathbf{z}_{ij}|^p$, $\|\mathbf{z}\|_\infty = \max_{i,j} |\mathbf{z}_{ij}|$.

A function $f : \mathbb{R}^{d \times n} \to \mathbb{R}^{d \times n}$ is *permutation equivariant* if for any permutation $\sigma \in S_n$, where $S_n$ is the symmetric group on $n$ elements, and for any $\mathbf{x} = (x_1, \ldots, x_n) \in \mathbb{R}^{d \times n}$, we have $f(x_{\sigma(1)}, \ldots, x_{\sigma(n)}) = \left(f_{\sigma(1)}(\mathbf{x}), \ldots, f_{\sigma(n)}(\mathbf{x})\right)$, where $f_i(\mathbf{x})$ denotes the $i$-th component of the output. We denote by $C^k(\mathbb{R}^d)$ the space of $k$-times continuously differentiable functions on $\mathbb{R}^d$. The space of continuous functions with compact support is denoted by $C_c(\mathbb{R}^d)$.

## 2 Problem Formulation

We briefly introduced the problem in Section 1, we restate the equations again for clarity. Let $\Omega \subset \mathbb{R}^d$. Consider a vector field $\mathcal{F} : \Omega \times \mathcal{P}(\Omega) \to \mathbb{R}^d$. The general mean-field behavior of interacting particles evolving on $\Omega$ is given by

$$\frac{dz}{dt} = \mathcal{F}(z, \mu), \ z(0) = z_0 \sim \mu_0, \tag{1}$$

where $z \in \Omega$ denotes the state of each particle, $\mu \in \mathcal{P}(\Omega)$ denotes the distribution of the particles, and $z_0$ is the initial state that is distributed according $\mu_0$, the initial distribution. The inter-particle interactions are modeled through $\mu$; specifically, the dynamics of each particle are influenced by the distribution of all other particles.

Corresponding to (1), the *continuity equation* describes the evolution of the distribution $\mu$:

$$\frac{\partial \mu}{\partial t} + \nabla_z \cdot (\mathcal{F}(z, \mu)\mu) = 0, \ \mu(0) = \mu_0. \tag{2}$$

For a finite final time $\tau > 0$, $\mu^{\mathcal{F}} : [0, \tau] \to \mathcal{P}(\Omega)$ denotes the solution of the continuity equation (2) over the time interval $[0, \tau]$.

In this paper, we propose to use transformers to approximate the map in (1) and the system (2). *However, transformers*

*are defined on sequences of vectors in $\mathbb{R}^d$, whereas the map $\mathcal{F}$ is defined on $\Omega \times \mathcal{P}(\Omega)$, an infinite-dimensional space.* Therefore, we consider a finite-dimensional approximation of (1) via a particle-level system. Specifically, consider a $n$-particle system where the state of each particle $i \in \{1, \ldots, n\}$ is given by $z_i \in \Omega$. We assume that each $z_i$ is independently sampled from the distribution $\mu \in \mathcal{P}(\Omega)$. Let $\mathbf{z} = (z_1, \ldots, z_n) \in \Omega^n$ denote the collection of particle states. The empirical distribution of the $n$-particle system is then given by

$$\nu_{\mathbf{z}}^n := \frac{1}{n} \sum_{i=1}^n \delta_{z_i} \in \mathcal{D}^n(\Omega). \tag{3}$$

The particle-level dynamics on $\mathbb{R}^d$ according to the map defined in (1), for a single particle $i$, can be written as

$$\dot{z}_i = \mathcal{F}(z_i, \nu_{\mathbf{z}}^n), \ z_i(0) \sim \mu_0 \tag{4}$$

Note that the collection of random variables $(z_i)$ is permutation equivariant because the joint distribution of $(z_i)$ is invariant under any permutation of the indices.

# 3 Numerical Simulations

In this section, we present experiments demonstrating that transformers can learn mean-field dynamics and generalize to cases involving more particles than those seen during training. We also compare the transformer with Graph Neural Networks (GNNs) and prior models from Pham & Warin (2023); Messenger & Bortz (2022).

## 3.1 Learning the Vector Field

Our first goal focuses on learning the vector field $\mathcal{F}$. Towards this, in this experiment, we use two datasets: first, a synthetic dataset generated from the Cucker-Smale model (Cucker & Smale, 2007), and second, real data of fish milling (Katz et al., 2021).

**Cucker-Smale Model** The first example that we consider is the well-studied $2d \times n$-dimensional Cucker-Smale (CS) equation that models consensus of a $n$-agent system (Cucker & Smale, 2007). For $d = 2$, he vector field $\mathcal{F} : \mathbb{R}^4 \times \mathcal{P}(\mathbb{R}^4) \to \mathbb{R}^4$ is given by

$$\mathcal{F}(x, v, \mu) = \begin{bmatrix} v \\ -\int_{\mathbb{R}^4} \phi(\|x - y\|)(v - u) d\mu(y, u) \end{bmatrix}, \\ \phi(r) = \frac{H}{(s^2 + r^2)^b}. \tag{5}$$

Where $x \in \mathbb{R}^2$ and $v \in \mathbb{R}^2$ denote the position and velocity of each agent, respectively. Here, $\phi$, a non-negative function, is the interaction potential that determines the inter-agent interaction, and $H, s, b$ are parameters (set to 1 here).

Next, we consider the $n$-dimensional (finite) approximation of the model above.

$$\begin{aligned} \frac{dx_i}{dt} &= v_i, \ 0 \leq i \leq n \\ \frac{dv_i}{dt} &= \frac{1}{n} \sum_{j=1}^n \phi(\|x_i - x_j\|)(v_j - v_i) \end{aligned} \tag{6}$$

To generate the data, we compute trajectories for 500 random initial conditions $(x_i(0), v_i(0))$, chosen uniformly at random from $\Omega = [0, 1] \times [0, 1]$. For each initial condition, we solve the differential equation (6) for $n = 20$ agents over a time horizon $[0, 100]$ using SciPy's `solve_ivp`. Hence, for each initial condition, we obtain position $x_i(t)$ and velocity $v_i(t)$, for each time step $t$.

**Fish Milling Model** The second example that we consider is the fish milling model (D'Orsogna et al., 2006) which is a version of the Cucker-Smale model. In contrast to the previous example, this example uses an actual experimental dataset (Katz et al., 2021). The dataset includes the position $x \in \mathbb{R}^2$, velocity $v \in \mathbb{R}^2$, and fish identifier for $n = 300$ fish over 5000 time steps. While the experiment tracks 300 fish, real-world data is incomplete at certain time points. Specifically, the positions of all fish are not available at all times, and as the fish move in a milling pattern, some fish disappear and reappear. When a fish reappears, it is assigned a new index, and its previous identity is lost. On average, only about 220 fish are observed at any given time. As a result, we have 5000 data points, but the number of particles varies at each time step.

**Data** Next, we will describe how we assemble the data from the Cucker-Smale and fish milling model for the ML architectures. We note that $\mathcal{F}$ maps $\begin{bmatrix} x & v \end{bmatrix}^T \mapsto \begin{bmatrix} \dot{x} & \dot{v} \end{bmatrix}^T$. Let $z(t) = \begin{bmatrix} x(t) & v(t) \end{bmatrix}^T$ and $\dot{z}(t) = \begin{bmatrix} \dot{x}(t) & \dot{v}(t) \end{bmatrix}^T$. From the simulations described above, we obtain the collection $\mathbf{z}(t) = (z_1(t), \ldots, z_n(t))$, where each $z_i(t) = \begin{bmatrix} x_i(t) & v_i(t) \end{bmatrix}^T$. To construct $\dot{z}(t)$, we use the centered-difference formula:

$$\dot{z}_i(t) \approx \frac{z_i(t + \Delta t) - z_i(t - \Delta t)}{2 \Delta t},$$

where $\Delta t$ is the time step. We assume that at each time step $t$, we may know $z(t)$ for only a subset of the particles; as is the case for the fish milling model. Moreover, we do not know the functional form of $\mathcal{F}(\mathbf{z}, \nu_{\mathbf{z}}^n)$. Let $I(t) := \{i : z_i(t), \dot{z}_i(t) \text{ are known}\}$. and we redefine $\mathbf{z}(t)$ to mean a collection of particle states for which we have data, and similarly for $\dot{\mathbf{z}}(t)$,

$$\mathbf{z}(t) = \{z_i(t) : i \in I(t)\} \text{ and } \dot{\mathbf{z}}(t) = \{\dot{z}_i(t) : i \in I(t)\}.$$

The data so obtained are split into an 80-20-20 split for training, validation, and testing.

| Model | Cucker-Smale | Fish Milling |
|---|---|---|
| **Transformer** | $(\mathbf{1.9 \pm 0.3}) \times \mathbf{10^{-6}}$ | $(\mathbf{2.2 \pm 0.0}) \times \mathbf{10^{-2}}$ |
| Cylindrical NN | $(1.1 \pm 0.4) \times 10^{-3}$ | $(2.7 \pm 0.0) \times 10^{-1}$ |
| GraphConv $m = 3$ | $(2.0 \pm 0.1) \times 10^{-4}$ | $(1.2 \pm 0.2) \times 10^{-1}$ |
| GraphConv $m = 20$ | $(4.1 \pm 3.6) \times 10^{-3}$ | $> 1$ |
| TransformerConv $m = 3$ | $(2.8 \pm 0.3) \times 10^{-5}$ | $(6.5 \pm 1.7) \times 10^{-2}$ |
| TransformerConv $m = 20$ | $(3.3 \pm 0.1) \times 10^{-6}$ | $(1.0 \pm 0.3) \times 10^{-1}$ |
| FNN | $(6.7 \pm 0.6) \times 10^{-6}$ | N/A |
| Kernel | $(3.0 \pm 0.1) \times 10^{-5}$ | N/A |

*Table 1.* Table showing the mean-squared error in approximating the map $F_n$ (14) for the different data and models.

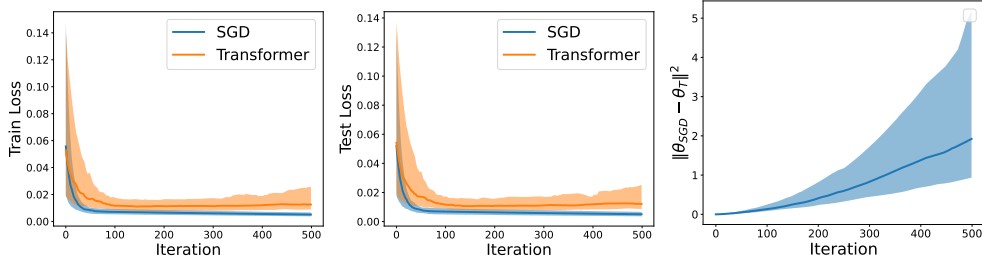

*Figure 1.* Figure comparing training a two-layer neural network using gradient descent to update the weights and using a transformer to update the weights. The solid line is the median value over 100 trials, while the shaded region is the interquartile range (25th-75th percentile). Left: evolution of the training error during training. Center: evolution of the test error during training. Right: difference between the parameters learned by gradient descent and the transformer.

Next, we will describe the ML architectures that we use to approximate $\mathcal{F}$. In particular, the transformer will be compared against four additional permutation equivariant baselines.

**Transformer** In Appendix D, we define a transformer network, $T : \mathbb{R}^{n \times d} \to \mathbb{R}^{n \times k}$ ($k$ not necessarily equal to $d$) At each time step $t$, we provide the transformer with input $\mathbf{z}(t)$ and have it predict $\dot{\mathbf{z}}(t)$. The model is trained using mean-squared error loss.

**Graph Neural Network (GNN)** Given a set of particles $Z(t)$, let $G_3(t)$ and $G_{20}(t)$ represent the graphs based on the three nearest neighbors and the twenty nearest neighbors, respectively. We use two graph neural networks (GNNs) that, at each time step $t$, take $(G_m(t), \mathbf{z}(t))$ as input, where $m$ is either 3 or 20, and aim to predict the node labels $\dot{\mathbf{z}}(t)$. This forms a graph regression task where the target size depends on the number of nodes in the graph.

We employ two common GNN architectures: the Graph Convolutional Network (GraphConv) (Morris et al., 2019) and TransformerConv (Shi et al., 2021). Note that, similar to GAT (Veličković et al., 2018), TransformerConv applies attention only to local neighborhoods.

**Other Baselines** The final three baselines we consider are: *cylindrical nets* from Pham & Warin (2023). Additionally, if

$|\dot{\mathbf{z}}(t)|$ is constant for all $t$ (i.e., for the CS data), we evaluate a fully connected feedforward net and kernel regression with sine, cosine, and polynomial basis. For both of these methods, we concatenate the vectors in $\mathbf{z}(t)$ and $\dot{\mathbf{z}}(t)$ to create the training data.

**Training Details and Hyperparameter Search** For each model and dataset, we conduct a hyperparameter search to optimize depth, width, and learning rate. We train the models using mini-batch Adam and a cosine annealing learning rate schedule. For kernel regression, we explore different numbers of basis functions for each type of basis function. See Appendix E for more training details.[1]

To account for the randomness introduced by initialization and training, we conduct five trials for each hyperparameter setting. The best hyperparameter configuration is selected based on performance on the validation data.

**Results** Table 1 presents the mean-squared error for the different methods, across the two data sets. We observe that the transformer has the lowest mean-squared error among all of the models. Therefore, we conclude (empirically) that transformers can effectively learn the vector field.

---

[1]Code can be found at: https://github.com/rsonthal/Mean-Field-Transformers

## 3.2 Simulating Mean-Field Dynamics

Our second goal is to approximate the solution to the continuity equation (2). We use two synthetic datasets: the Cucker-Smale dataset from Section 3.1, and the training dynamics for a two-layer neural network (Mei et al., 2019).

**Cucker-Smale** After obtaining a transformer $T$ that approximates the vector field $\mathcal{F}$ in (5) (as described in Section 3.1), we solve the differential equation (6) twice, using SciPy's `solve_ivp` function, for a new set of initial condition. First, with the true vector field $\mathcal{F}$, and second, using the transformer $T$ in lieu of $\mathcal{F}$. We note that solving (2) for Dirac valued initial distributions, $\delta_{z_i(0)}$, is equivalent to solving (6) for initial condition $z_i(0)$.

**Training 2-Layer Neural Network** Consider a two-layer network $f(x) = \sum_{i=1}^{n} a_i \sigma(x^T w_i)$. Let $\theta_i = (a_i, w_i)$ be the parameters (the states of this model) and $\boldsymbol{\Theta} = (\theta_1, \ldots, \theta_n)$. In this model, we treat each $\theta_i$ as a particle, with its distribution evolving according to the following continuity equation, as derived by Mei et al. (2019),

$$\dot{\mu} = 2\xi(t)\nabla_\theta \cdot (\mu\nabla_\theta\Psi(\theta, \mu)). \tag{7}$$

Here, $\xi(t)$ depends on the learning rate schedule and

$$\Psi(\theta, \mu) := -\mathbb{E}_{x,y}\left[ya\sigma(x^T w)\right]$$
$$+ \int \mathbb{E}_{x,y}\left[a\hat{a}\sigma(x^T w)\sigma(x^T \hat{w})\right] d\mu(\hat{a}, \hat{w}).$$

Here, the vector field we wish to approximate is $\mathcal{F}(\theta, \mu) = 2\xi(t)\nabla_\theta\Psi(\theta, \mu)$. To generate the data, let $X$ be a $d$ times 1000 dimensional matrix with i.i.d. Gaussian entries. Let $f$ be a fixed two-layer teacher network with sigmoid activation ($\sigma$) and let $Y = f(X)$. We then initialize 50 two-layer networks with $n$ hidden nodes, and train them for 500 epochs using the data $X, Y$. We use the MSE loss and gradient descent to train the network. This gives us 2500 training data points $(\boldsymbol{\Theta}, \dot{\boldsymbol{\Theta}})$. We set the hidden layer with $n = 100$ and use an input dimension of $d = 10$.

Once the transformer is trained, we test it as follows: We reinitialize a two-layer network, providing its weights as input to the transformer. The transformer's outputs are then treated as gradients, which we use to perform gradient descent. Finally, we compare the results with the dynamics obtained from performing true gradient descent.

**Results** Figure 3 shows the evolution of 30 particles using the Cucker-Smale equations (6) along with the transformer approximation. Notably, the dynamics obtained using the transformer tracks the true dynamics near exactly. Figure 2 plots the 2-norm between the positions coordinates $x, y$ and velocities $u, v$. The figure indicates that the error is generally quite small ($< 10^{-4}$), although it increases over time. This increase appears to be linear for the position coordinates,

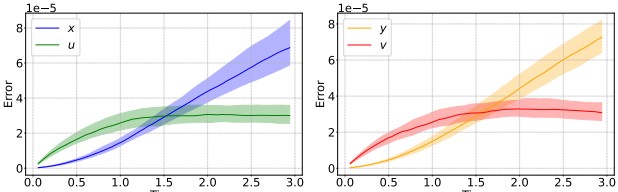

*Figure 2.* Figure comparing the true dynamics of the Cucker-Smaler model versus those obtained from a transformer. The solid line is the median value over 100 trials, while the shaded region is the interquartile range (25th-75th percentile).

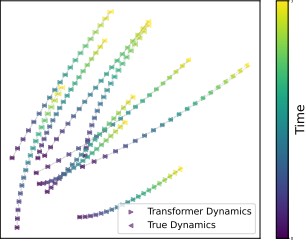

*Figure 3.* Figure showing the trajectories of ten particles computed for the Cucker-Smale model using the true $\mathcal{F}$ versus the transformer in lieu of $\mathcal{F}$.

while the error in the velocity seems to plateau and even decrease slightly. Additionally, the initial interquartile range is small, but it grows over time.

Finally, we note that we trained the model with 20 particles but simulated the model with 30 particles. Hence we see that the model can generalize to more particles than that seen during training. This implies that the transformer is able to learn the interaction kernel between the particles.

Next we simulate training a two-layer neural network using a Transformer. Figure 1 illustrates the training and test loss as we train the network. The blue line represents the model trained using GD, while the orange line corresponds to the model trained with the transformer. Note that the transformer model does not compute any gradients. We trained models with one hundred different random initializations. The solid lines indicate the median and the shaded regions represent the interquartile range. The right-most plot in Figure 1 shows the Frobenius norm of the difference between the parameters learned using GD and the transformer. The figure demonstrates that the maximum norm of the difference is at most $5 \times 10^{-5}$, even after 500 iterations. Therefore, we observe that the transformer has learned to approximate the dynamics of GD for a two-layer network.

## 4 Theoretical Results

Our experiments demonstrate that the transformer can approximate the finite-dimensional particle dynamics (4). In

this section, we will prove that the infinite-dimensional vector field $\mathcal{F}$ (1) can be approximated by the expected output of a transformer, which we shall denote $\mathcal{T}_n(x, \mu)$. Further, we show that the solutions to the continuity equation (2) can be approximated by the solutions to the following *approximate continuity equation*, defined using $\mathcal{T}_n$,

$$\frac{\partial \mu}{\partial t} + \nabla_z \cdot (\mathcal{T}_n(z, \mu)\mu) = 0, \ \ \mu(0) = \mu_0. \tag{8}$$

### 4.1 Lifting Transformers to the Space of Measures

Traditionally, transformers are defined on sequences of vectors in $\mathbb{R}^d$. However, the map we wish to approximate, $\mathcal{F}$, is defined on $\Omega \times \mathcal{P}(\Omega)$. Therefore, we lift the standard transformer to act on $\Omega \times \mathcal{P}(\Omega)$ via an expectation operation. This will allow us to lift any transformer $T : \Omega^{n+1} \to \mathbb{R}^{(n+1) \times d}$ to a model $\mathcal{T}_n : \Omega \times \mathcal{P}(\Omega) \to \mathbb{R}^d$.

**Definition 4.1** (Expected Transformer). Given a transformer $T : \Omega^{n+1} \to \mathbb{R}^{(n+1) \times d}$ and a prescribed sequence length $n$, define the expected transformer $\mathcal{T}_n : \Omega \times \mathcal{P}(\Omega) \to \mathbb{R}^d$ by

$$\mathcal{T}_n(x, \mu) := \mathbb{E}_{\mathbf{z} \sim \mu^{\otimes n}} \left[ (T([x; \mathbf{z}]))_1 \right], \tag{9}$$

$$= \int_{\Omega^n} (T([x; z_1, \ldots, z_n]))_1 \ d\mu(z_1) \cdots d\mu(z_n),$$

where $[x; \mathbf{z}]$ denotes the concatenation of $x$ and $\mathbf{z} = (z_1, \ldots, z_n)$ to form an input sequence of length $n + 1$, and $(T([x; \mathbf{z}]))_1$ denotes the first output vector.

Some prior works (Geshkovski et al., 2023; Furuya et al., 2025) have defined transformers $\hat{T} : \Omega \times \mathcal{P}(\Omega) \to \mathbb{R}^d$ through a continuous version of self-attention $\Gamma$. For $x \in \Omega$ and $\mu \in \mathcal{P}(\Omega)$, $\Gamma$ is defined as

$$\Gamma(x, \mu) := x + \frac{1}{Z(x, \mu)} \int_\Omega \text{Att}([x; y]) \ d\mu(y), \tag{10}$$

where $Z(x, \mu)$ is a normalization factor and $\text{Att}$ is the attention layer (D.1). Then the transformer $\hat{T}$ in Geshkovski et al. (2023); Furuya et al. (2025) is defined as

$$\hat{T}(x, \mu) := \text{FC}_{\xi_L} \circ \Gamma_{\theta_L} \circ \cdots \circ \text{FC}_{\xi_1} \circ \Gamma_{\theta_1}(x), \tag{11}$$

where $\Gamma_{\theta_j}$ and $\text{FC}_{\xi_j}$ are attention and feed-forward layers with parameters $\theta_j$ and $\xi_j$, respectively.

### 4.2 Assumptions

To state our result, we require some assumptions on the map $\mathcal{F}$ in (1). The key assumption we make is that $\mathcal{F}$ is Lipschitz continuous with respect to $\mu$, the probability measure. To formalize this, we require a metric on the space of probability measures $\mathcal{P}(\Omega)$. A commonly used metric is the $p$-Wasserstein distance.

**Definition 4.2** (1-Wasserstein Distance). Given two probability measures $\mu, \nu \in \mathcal{P}_p(\Omega)$ on a metric space $(\Omega, \| \cdot \|_2)$, where $d$ is the metric on $\Omega$, the 1-Wasserstein distance between $\mu$ and $\nu$ is defined as

$$\mathcal{W}_1(\mu, \nu) = \inf_{\gamma \in \Pi(\mu, \nu)} \int_{\Omega \times \Omega} \|x - y\|_2 d\gamma(x, y), \tag{12}$$

where $\Pi(\mu, \nu)$ denotes the set of all couplings (transport plans) $\gamma$ on $\Omega \times \Omega$ with marginals $\mu$ and $\nu$.

We now state the main assumptions required for our analysis:

**Assumption 4.3** (Regularity and Growth Conditions). Assume that the vector field $\mathcal{F} : \Omega \times \mathcal{P}_p(\Omega) \to \mathbb{R}^d$ satisfies the following conditions:

a) (**Lipschitz Continuity**) There exists a constant $\mathscr{L}$ such that for all $x, y \in \Omega$ and $\mu, \nu \in \mathcal{P}_p(\Omega)$,

$$\|\mathcal{F}(x, \mu) - \mathcal{F}(y, \nu)\|_2 \leq \mathscr{L} \left( \|x - y\|_2 + \mathcal{W}_1(\mu, \nu) \right).$$

This condition implies the following.

b) (**Linear Growth**) There exists a constant $\mathscr{M} > 0$ such that for all $x \in \Omega$ and $\mu \in \mathcal{P}_p(\Omega)$,

$$\|\mathcal{F}(x, \mu)\|_2 \leq \mathscr{M} \left( 1 + \|x\|_2 + M_1(\mu) \right),$$

where $M_1(\mu) := \int_\Omega |y| d\mu(y)$ is the first moment of $\mu$.

These assumptions are standard in the analysis of mean-field models and differential equations in general and are needed to guarantee the uniqueness of solutions.

*Remark* 4.4 (Lipschitz Implies Linear Growth). Since $\Omega$ is compact, Assumption 4.3a) implies Assumption 4.3b). However, in some cases, we only need the weaker assumption of Linear Growth, hence we explicitly state it.

*Remark* 4.5 (Example Models). These assumptions are satisfied by the Cucker-Smale model as well as the training 2-layer neural network model. Specifically, Theorem 2 of Piccoli et al. (2009) shows that Cucker-Smale model satisfies Assumption 4.3a) and Lemma 2 of Mei et al. (2019) shows that the training 2-layer neural network model (7) also satisfies Assumption 4.3a)

To establish our approximation results for functions $\mathcal{H} : \Omega \times \mathcal{P}(\Omega) \to \mathbb{R}^d$, we define the following norm.

**Definition 4.6.** Given a function $\mathcal{H} : \Omega \times \mathcal{P}(\Omega) \to \mathbb{R}^d$, we define its norm by

$$\|\mathcal{H}\|_* := \sup_{x \in \Omega} \sup_{\mu \in \mathcal{P}(\Omega)} \|\mathcal{H}(x, \mu)\|_2. \tag{13}$$

### 4.3 Approximating the Vector Field $\mathcal{F}$

To approximate the vector field $\mathcal{F}$, we define, for a fixed $n$, the finite-dimensional map $F_n : \Omega^n \to \mathbb{R}^{d \times n}$ as

$$F_n(\mathbf{z}) := \begin{bmatrix} \mathcal{F}(z_1, \nu_{\mathbf{z}}^n) & \ldots & \mathcal{F}(z_n, \nu_{\mathbf{z}}^n) \end{bmatrix} \tag{14}$$

We now state our main result regarding the universal approximation of the mean-field vector field $\mathcal{F}$ by the expected transformer $\mathcal{T}_n$.

**Theorem 4.7** (Universal Approximation). *Let $\Omega \subset \mathbb{R}^d$ be a compact set containing 0. Let $\mathcal{F} : \Omega \times \mathcal{P}(\Omega) \to \mathbb{R}^d$ satisfy Assumption 4.3a) for a given $p$. Given a transformer $T : \Omega^{n+1} \to \mathbb{R}^{(n+1)\times d}$ let*

$$\mathcal{E} := \sup_{\mathbf{z}\in\Omega^{n+1}} \|T(\mathbf{z}) - F_{n+1}(\mathbf{z})\|_2 \tag{15}$$

*Then, for $q > p$ there exists a constant $C(p, q, d)$, depending only on $p$, $q$, and $d$, such that for all $n \geq 1$, the corresponding continuum version $\mathcal{T}_n : \Omega \times \mathcal{P}(\Omega) \to \mathbb{R}^d$ (9) satisfies*

$$\|\mathcal{T}_n - \mathcal{F}\|_* \leq \mathcal{E} + \mathscr{L} diam(\Omega)^p \left( \frac{1}{n+1} + CG(n,p,q) \right),$$

$$G(n,p,q) = \begin{cases} \frac{1}{\sqrt{n}} & p > d/2,\ q \neq 2p \\ \frac{1}{\sqrt{n}}\log(n+1) & p = d/2,\ q \neq 2p \\ \frac{1}{\sqrt[d]{n^p}} & p < d/2,\ q \neq \frac{d}{d-p} \end{cases} \tag{16}$$

*Remark* 4.8. In the Theorem 4.7 above, we observe that the approximation of infinite-dimensional maps $\mathcal{F}$ by finite-dimensional transformers $T$ depends on two key quantities. First, it depends on how well $T$ approximates the finite-dimensional map $F_n$. This corresponds to the $\mathcal{E}$ term in the bound (15). Due to universal approximation results for transformers, this term can be made arbitrarily small. For example, see Theorem 4.3 in Alberti et al. (2023).

Second, the approximation depends on the convergence rate of $\mathcal{W}_1(\mu, \nu_\mathbf{z}^n)$, which, by Theorem 1 of Fournier & Guillin (2015) is tight and depends on $n$, $p$, $q$, and $d$. Furthermore, we observe that the stronger the regularity of the map $\mathcal{F}$ (i.e., larger the constant $p$), the better the approximation. The best rates are obtained for $p = \lfloor \frac{d}{2} + 1 \rfloor$. In which case the rate becomes independent of the dimension $d$. Consequently, if we use longer sequences, we obtain an improved approximation of the vector field $\mathcal{F}$.

We empirically verify Theorem 4.7 using the transformer trained for the Cucker-Smale model in Section 3.1. We test five $(x, y) \in [0.1]^2$ values and vary $(u, v)$ over an $11 \times 11$ grid. Using Gibbs sampling, we approximate the expected transformer and compute the error with the true vector field (5). Figure 4 shows the heat maps, with a maximum error of 0.025, while the bound from Theorem 4.7 is at least 0.05.

**Comparison with Result From (Furuya et al., 2025):** The concurrent work (Furuya et al., 2025) proves the following approximation result for continuous maps $\mathcal{F}$ by the continuum version of the transformer $\hat{T}$ (11).

**Theorem 4.9.** *Let $\Omega \subset \mathbb{R}^d$ be a compact set and $F^* : \Omega \times \mathcal{P}(\mathbb{R}^d) \to \mathbb{R}^d$ be continuous, where $\mathcal{P}(\mathbb{R}^d)$ is endowed with the weak\* topology. Then for all $\varepsilon > 0$, there exist $l$ and parameters $(\theta_j, \xi_j)_{j=1}^l$ such that*

$$\|\hat{T}(x, \mu) - F^*(x, \mu)\|_2 \leq \varepsilon, \quad \forall (x, \mu) \in \Omega \times \mathcal{P}(\mathbb{R}^d)$$

*where the parameters $\theta_j, \xi_j$ depend on the dimension $d$.*

To compare with Theorem 4.9, we state the following corollary to Theorem 4.7.

**Corollary 4.10.** *Let $\varepsilon > 0$ and $n \geq 1$. Let $\Omega \subset \mathbb{R}^d$ be a compact set containing 0. Let $\mathcal{F} : \Omega \times \mathcal{P}(\Omega) \to \mathbb{R}^d$ satisfy Assumption 4.3a) for a given $p$. Then there exists a transformer $T$ with depth $\Theta(1)$, one attention layer with width $\Theta(d)$ such that the expected transformer $\mathcal{T}_n$ satisfies (16) with $\mathcal{E} = \varepsilon$.*

While Corollary 4.10 and Theorem 4.9 are about two different models, they share notable similarities while exhibiting key differences. Both results feature $\Theta(d)$ width for the attention layers, independent of $\varepsilon$ and $n$, and neither provides bounds for the width of feedforward layers. However, there are key differences. Our work leverages prior results to establish a bound on network depth, which Furuya et al. (2025) does not. Moreover, we note that providing a bound on the width of the feedforward network, in our case, is straightforward, owing to recent developments that provide bounds on both width and depth (Augustine, 2024).

Further, we provide detailed error rates that depend on the length of the sequence $n$ that the transformer is trained on. However, we note that Furuya et al. (2025) impose weaker assumptions for the map $\mathcal{F}$.

### 4.4 Approximating the Mean Field Dynamics

We build upon our previous approximation results to prove that the solution $\mu^{\mathcal{T}_n}(t)$ to the approximate continuity equation (8) approximates the solution $\mu^{\mathcal{F}}(t)$ to continuity equation (2). To formalize this, we first introduce an appropriate notion of a solution to the continuity equation.

**Definition 4.11.** Let $\mu \in C([0, \tau]; \mathcal{P}_p(\mathbb{R}^d))$ be a measure-valued function, is called a *Lagrangian solution* of the continuity (2) if there exists $X : [0, \tau] \times \mathbb{R}^d \to \mathbb{R}^d$, referred to as the *flow ma*p, satisfies

$$X(t, x) = x + \int_0^t \mathcal{F}(X(s, x), \mu(s))ds, \tag{17}$$

for all $x \in \mathbb{R}^d$ and $\mu(t) = X(t, \cdot)_\# \mu_0$ for all $t \in [0, \tau]$.

**Existence and Uniqueness of Solutions** Under Assumption 4.3a), it is known that there is a unique Lagrangian solution corresponding to (2), see Proposition 4.8 in Cavagnari et al. (2022). As stated in Remark 4.5 the numerical examples considered in Section 3 satisfy this assumption.

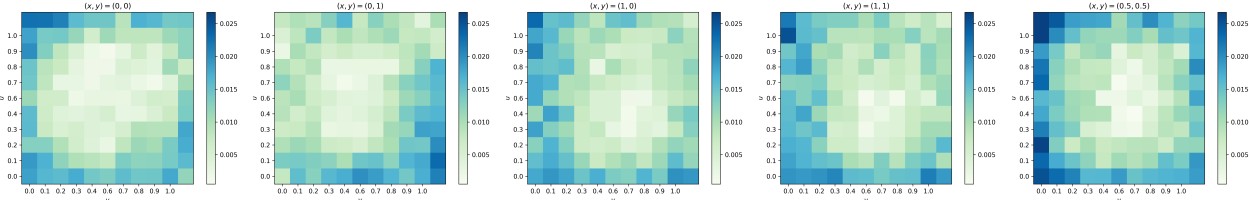

*Figure 4.* Figure shows the error $\|\mathcal{T}_n - \mathcal{F}\|_*$ for the CS model. Here $(x, y)$ is held fixed while $(u, v)$ is varied in a $11 \times 11$ grid.

However, to ensure that solutions of (2) can be approximated, we also need existence of unique solutions for the approximating continuity equation (8). Unfortunately, the expected transformer might not be globally Lipschitz as required in Assumption *4.3a)* since $T[x, \mathbf{z}]$ in (9) might not be globally Lipschitz (Kim et al., 2021). Hence, the continuity equation (8) with the expected transformer as the vector field may not have a unique Lagrangian solution in general.

The goal of the following theorem is to show that if Assumption *4.3b)* holds locally on a sufficiently large set, then we can prove existence of a unique solution to (8), even if the expected transformer is only locally Lipschitz. This will be used later to show the ability of the expected transformer to approximate solutions of the continuity equation (2), using its universal approximation property.

**Theorem 4.12.** *Suppose $\mu_0 \in \mathcal{P}_c(\Omega)$ is such that the support of $\mu_0$ is contained within $B_R(0) \subset \Omega$, for some $R > 0$. Additionally, let $K$ satisfy $K > (R + 2\mathcal{M}\tau)e^{3\mathcal{M}\tau}$. Furthermore, assume that*

$$\|\mathcal{T}_n(x, \mu)\|_2 \leq \mathcal{M}\left(1 + \|x\|_2 + M_1(\mu)\right), \qquad (18)$$

*for all $x \in B_K(0)$ and all $\mu$ with support in $B_K(0)$. Then, there exists a unique Lagrangian solution to the approximate continuity equation (8) such that $\operatorname{supp}\mu^{\mathcal{T}}(t) \subseteq B_{C_t}(0)$ for all $t \in [0, \tau]$, where $C_t = (R + 2\mathcal{M}t)e^{3\mathcal{M}t}$.*

*Remark* 4.13. In the proof of Theorem 4.14 it will be shown that the bound (18) on the expected transformer directly follows from Assumption *4.3a)* on $\mathcal{F}$ whenever $\mathcal{T}_n$ is close enough to $\mathcal{F}$ in the uniform norm.

We are now ready to state our main theorem regarding the approximation of mean-field dynamics using transformers.

**Theorem 4.14** (Mean Field Dynamics Approximation Using Transformers)**.** *Let $\varepsilon > 0$ be small enough and $n \geq 1$. Suppose that $\mathcal{F}$ satisfies Assumption 4.3a) for some $p$. Assume that the support of $\mu_0$ is contained within $B_R(0) \subset \Omega$, for some $R > 0$, and let $K \in \mathbb{R}$ be such that $K > (R + 2\mathcal{M}\tau)e^{3\mathcal{M}\tau}$. If the transformer $\mathcal{T}_n$ satisfies the condition*

$$\|\mathcal{T}_n(x, \mu) - \mathcal{F}(x, \mu)\|_* < \varepsilon.$$

*for all $z \in B_K(0)$ and $\mu \in \mathcal{P}(B_K(0))$. Then we have that*

$$\mathcal{W}_1(\mu^{\mathcal{F}}(t), \mu^{\mathcal{T}_n}(t)) \leq \varepsilon t \exp(2\mathcal{L}t) \qquad (19)$$

*where $\mu^{\mathcal{F}}$ and $\mu^{\mathcal{T}_n}$ are the solutions to (2) and (8), respectively, and the estimate (19) is independent of $\mu_0$.*

*Remark* 4.15. In essence, Theorem 4.14 states that by choosing a large enough ball of radius $R$ that covers the support of $\mu_0$, and selecting and appropriate $K$, as a function of $R$, final time $\tau$, and regularity constant $\mathcal{M}$, we can ensure that if $\mathcal{T}_n$ approximates $\mathcal{F}$ on the ball of radius $K$, then $\mathcal{T}_n$ can be used to simulate the dynamics (2) over the time interval $[0, \tau]$. We observe that the error bound (19) grows exponentially. Therefore, a small approximation error $\delta$ for the vector field $\mathcal{F}$ implies a small approximation error (19) for the solution of the continuity equation over $[0, \tau]$. However, the bound (19) also depends on the regularity of $\mathcal{F}$, namely $p$ and $\mathcal{L}$. Therefore, the more regular the vector-field $\mathcal{F}$, i.e., larger $p$ and smaller $\mathcal{L}$, the better the bound (19).

We can combine Theorem 4.7 and Theorem 4.14 to obtain the following corollary.

**Corollary 4.16.** *Suppose $\mathcal{F}$ satisfies Assumption 4.3a) for some $p$. Let the support of $\mu_0$ be contained within $B_R(0) \subset \Omega$, for some $R > 0$, and let $K \in \mathbb{R}$ be such that $K > (R + 2\mathcal{M}\tau)e^{3\mathcal{M}\tau}$. Given a transformer $T : \Omega^{n+1} \to \mathbb{R}^{(n+1)\times d}$ let $\mathcal{E} := \sup_{\mathbf{z} \in \Omega^{n+1}} \|T(\mathbf{z}) - F_{n+1}(\mathbf{z})\|_2$. If $\mathcal{E}$ is small enough and $n \in \mathbb{Z}_+$ is large enough, then*

$$W_1(\mu^{\mathcal{F}}(t), \mu^{\mathcal{T}_n}(t)) < 2(\mathcal{E} + \delta(n, K))t \exp(2^p \mathcal{L}t)$$

*where $\delta(n, K) = \mathcal{L}(2K)^p \left( \frac{1}{n^{\frac{q-p}{q}}} G(n, p, q) \right)$, where $G(n, p, q)$ is as per (16).*

## 5 Related Works

In this section, we provide a more in-depth comparison with similar recent prior work Furuya et al. (2025); Kratsios & Furuya (2025). We highlight key differences below.

**Role of Measure and Approximation Target:** In this work the measure $\mu$ is an input argument representing the state distribution in a mean-field system, directly influencing the vector field $\mathcal{F}(z, \mu)$. Specifically, this paper targets the approximation of the vector field $\mathcal{F}(z, \mu)$ governing mean-field dynamics and the subsequent approximation of the dynamical system's evolution (solution to the continuity

equation). We require the target vector field $\mathcal{F}(z, \mu)$ to be Lipschitz continuous w.r.t. spatial and measure arguments (using Wasserstein distance). This directly models physical or biological system interactions. A key strength is its applicability to general Borel probability measures $\mathcal{P}(\Omega)$ on a compact set $\Omega$.

Furuya et al. (2025) also has $\mu$ as input and aim to approximate general continuous in-context mappings $\Lambda^*(\mu, x)$. They require $\Lambda^*$ to be continuous w.r.t. weak-star topology (plus Lipschitz conditions on contexts for masked case). Their focus is broad representational power.

For Kratsios & Furuya (2025) the measure $\mu$ is an input argument to the target function $f(\mu, x)$. However, it's restricted to the specific class of Permutation-Invariant Contexts within a geometrically constrained domain, aimed at analyzing general in-context function approximation.

**Map Definition and Transformer:** Our work defines the map $\mu \mapsto \mathcal{T}_n$ (Measure to Vector Field) using the Expected Transformer, derived by taking the expectation of a standard, finite-sequence transformer $T$. This provides a practical link between standard architectures and measure-theoretic inputs.

Furuya et al. (2025) define transformers directly on the space of probability measures using a measure-theoretic formulation with continuous attention layers. Kratsios & Furuya (2025) defines the map from finite vector spaces to finite vector spaces. However, the input and outputs are interpreted as measures on discrete sets.

**Guarantees:** This work provides quantitative bounds on the vector field approximation error that explicitly show convergence as the number of particles increases, linking the error to the quality of the underlying finite transformer. Furthermore, it connects this to the approximation of the system's dynamics via Gronwall's lemma.

Kratsios & Furuya (2025) provides quantitative probabilistic bounds on the output approximation error, dependent on the target function's modulus of continuity and the domain's geometry, focusing on the network size needed for a given precision. While Furuya et al. (2025) show a single transformer architecture (with fixed dimensions/heads) works uniformly for an arbitrary number of input tokens (even infinite) for a given precision $\varepsilon$.

## 6 Conclusion

This paper demonstrated the efficacy of transformer architectures in approximating the mean-field dynamics of interacting particle systems. We showed that finite-dimensional transformer models can be lifted to approximate the infinite-dimensional mean-field dynamics. Through theoretical results and numerical simulations, we established that transformers can be powerful tools for modeling and learning the collective behavior of particle systems.

## Acknowledgment

Shiba Biswal acknowledges partial support from the Laboratory Directed Research and Development program of Los Alamos National Laboratory (LANL) under Projects No. 20230338ER and No. 20250638DI, and from the U.S. Department of Energy/Office of Science Advanced Scientific Computing Research Program.

## Impact Statement

This paper presents work whose goal is to advance the field of Machine Learning. There are many potential societal consequences of our work, none which we feel must be specifically highlighted here.

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

# A  Proof of Theorem 4.7

**Theorem 4.7** (Universal Approximation). *Let $\Omega \subset \mathbb{R}^d$ be a compact set containing $0$. Let $\mathcal{F} : \Omega \times \mathcal{P}(\Omega) \to \mathbb{R}^d$ satisfy Assumption 4.3a) for a given $p$. Given a transformer $T : \Omega^{n+1} \to \mathbb{R}^{(n+1) \times d}$ let*

$$\mathcal{E} := \sup_{\mathbf{z} \in \Omega^{n+1}} \|T(\mathbf{z}) - F_{n+1}(\mathbf{z})\|_2 \tag{15}$$

*Then, for $q > p$ there exists a constant $C(p, q, d)$, depending only on $p$, $q$, and $d$, such that for all $n \geq 1$, the corresponding continuum version $\mathcal{T}_n : \Omega \times \mathcal{P}(\Omega) \to \mathbb{R}^d$ (9) satisfies*

$$\|\mathcal{T}_n - \mathcal{F}\|_* \leq \mathcal{E} + \mathscr{L} \, diam(\Omega)^p \left( \frac{1}{n+1} + CG(n, p, q) \right),$$

$$G(n, p, q) = \begin{cases} \frac{1}{\sqrt{n}} & p > d/2, \ q \neq 2p \\ \frac{1}{\sqrt{n}} \log(n+1) & p = d/2, \ q \neq 2p \\ \frac{1}{\sqrt[d]{n^p}} & p < d/2, \ q \neq \frac{d}{d-p} \end{cases} \tag{16}$$

*Proof.*

$$\begin{aligned}
\|\mathcal{F} - \mathcal{T}_n\|_* &= \sup_\mu \sup_x \left\| \mathcal{F}(x, \mu) - \int_{\Omega^n} (T(x, \mathbf{z}))_1 \, d\mu^{\otimes n}(\mathbf{z}) \right\|_2 \\
&\leq \sup_\mu \sup_x \int_{\Omega^n} \|\mathcal{F}(x, \mu) - (T(x, \mathbf{z}))_1\|_2 \, d\mu^{\otimes n}(\mathbf{z}) \\
&= \sup_\mu \sup_x \int_{\Omega^n} \|\mathcal{F}(x, \mu) - \mathcal{F}(x, \nu_{\mathbf{z}}^n) + \mathcal{F}(x, \nu_{\mathbf{z}}^n) - (T(x, \mathbf{z}))_1\|_2 \, d\mu^{\otimes n}(\mathbf{z}) \\
&\leq \sup_\mu \sup_x \int_{\Omega^n} \|\mathcal{F}(x, \mu) - \mathcal{F}(x, \nu_{\mathbf{z}}^n)\|_2 \, d\mu^{\otimes n}(\mathbf{z}) \\
&\quad + \sup_\mu \sup_x \int_{\Omega^n} \|\mathcal{F}(x, \nu_{\mathbf{z}}^n) - (T(x, \mathbf{z}))_1\|_2 \, d\mu^{\otimes n}(\mathbf{z}) \qquad (*)
\end{aligned}$$

The second inequality follows from the standard triangle inequality. The first integral on the RHS can be bounded from above as,

$$\begin{aligned}
\sup_\mu \sup_x & \int_{\Omega^n} \|\mathcal{F}(x, \mu) - \mathcal{F}(x, \nu_{\mathbf{z}}^n)\|_2 \, d\mu^{\otimes n}(\mathbf{z}) \\
&\leq \sup_\mu \int_{\Omega^n} \sup_x \|\mathcal{F}(x, \mu) - \mathcal{F}(x, \nu_{\mathbf{z}}^n)\|_2 \, d\mu^{\otimes n}(\mathbf{z}) \\
&= \sup_\mu \int_{\Omega^n} \mathscr{L} \|\mu - \nu_{\mathbf{z}}^n\|_{\mathcal{W}_p} \, d\mu^{\otimes n}(\mathbf{z}) \\
&= \mathscr{L} \sup_\mu \mathbb{E}_{\mathbf{z} \sim \mu^{\otimes n}} [\mathcal{W}_p(\mu, \nu_{\mathbf{z}}^n)] \qquad (**)
\end{aligned}$$

Recall that $M_q(\mu)$ denotes the $q$-moment of $\mu$ i.e. $M_q(\mu) := \int_\Omega |x|^q d\mu(x)$, then as per Theorem 1 of (Fournier & Guillin, 2015) there exists a constant $C(p, q, d)$ (a function of $p, q, d$) such that, $\mathbb{E}_{\mathbf{z} \sim \mu^{\otimes n}} [\mathcal{W}_p(\mu, \nu_{\mathbf{z}}^n)]$ from $(**)$ can be bounded from above by $C M_q^{p/q}(\mu) G(n, p, q)$. We obtain:

$$\begin{aligned}
\sup_\mu \sup_x \int_{\Omega^n} \|\mathcal{F}(x, \mu) - \mathcal{F}(x, \nu_{\mathbf{z}}^n)\|_2 \, d\mu^{\otimes n}(\mathbf{z}) &\leq \mathscr{L} \sup_\mu C M_q^{p/q}(\mu) G(n, p, q) \\
&\leq \mathscr{L} C \, \mathrm{diam}(\Omega)^p G(n, p, q), \tag{20}
\end{aligned}$$

where we have used the fact that $\mu$ is a probability measure on $\Omega$.

Next, we obtain an upper bound for the second integral in $(*)$.

$$\sup_{\mu} \sup_{x} \int_{\Omega^n} \|\mathcal{F}(x, \nu_{\mathbf{z}}^n) - (T(x, \mathbf{z}))_1\|_2 \, d\mu^{\otimes n}(\mathbf{z})$$

$$\leq \sup_{\mu} \int_{\Omega^n} \sup_{x} \|\mathcal{F}(x, \nu_{\mathbf{z}}^n) - (F_{n+1}(x, z_1, \ldots, z_n))_1\|_2 \, d\mu^{\otimes n}(\mathbf{z})$$

$$+ \sup_{\mu} \int_{\Omega^n} \sup_{x} \|(F_{n+1}(x, z_1, \ldots, z_n))_1 - (T(x, \mathbf{z}))_1\|_2 \, d\mu^{\otimes n}(\mathbf{z}) \qquad (***)$$

Consider the first term in the expression above

$$\sup_{\mu} \int_{\Omega^n} \sup_{x} \|\mathcal{F}(x, \nu_{\mathbf{z}}^n) - (F_{n+1}(x, z_1, \ldots, z_n))_1\|_2 \, d\mu^{\otimes n}(\mathbf{z})$$

$$= \sup_{\mu} \int_{\Omega^n} \sup_{x} \left\|\mathcal{F}(x, \nu_{\mathbf{z}}^n) - \mathcal{F}\left(x, \nu_{(x,\mathbf{z})}^{n+1}\right)\right\|_2 \, d\mu^{\otimes n}(\mathbf{z})$$

$$\leq \sup_{\mu} \int_{\Omega^n} \mathscr{L} \mathcal{W}_1\left(\nu_{\mathbf{z}}^n, \nu_{(x,\mathbf{z})}^{n+1}\right) \, d\mu^{\otimes n}(\mathbf{z})$$

$$\leq \sup_{\mu} \int_{\Omega^n} \mathscr{L} \operatorname{diam}(\Omega)^p \frac{1}{n+1} d\mu^{\otimes n}(\mathbf{z})$$

$$= \mathscr{L} \operatorname{diam}(\Omega)^p \frac{1}{n+1} \tag{21}$$

Since we have assumed (15), the second term in $(***)$ evaluates to

$$\sup_{\mu} \int_{\Omega^n} \sup_{x} \|(F_{n+1}(x, z_1, \ldots, z_n))_1 - (T(x, \mathbf{z}))_1\|_2 \, d\mu^{\otimes n}(\mathbf{z}) = \mathcal{E}. \tag{22}$$

Putting together (20), (21), and (22), we get that

$$\|\mathcal{T}_n - \mathcal{F}\|_* \leq \mathscr{L} \operatorname{diam}(\Omega)^p \left(CG(n, p, q) + \frac{1}{n+1}\right) + \mathcal{E}.$$

$\square$

# B    Proof of Theorem 4.12

**Proposition B.1.** *Suppose $\mu_0 \in \mathcal{P}(\Omega)$ is such that the support of $\mu_0$ lies in $B_R(0)$, for some $R > 0$, and that there exists a Lagrangian solution $\mu^{\mathcal{F}} \in C([0, \tau]; \mathcal{P}_p(\mathbb{R}^d))$ of (2). Additionally, suppose that $\mathcal{F}$ satisfies Assumption 4.3b). Then the solution satisfies,*

$$\operatorname{supp} \mu^{\mathcal{F}}(t) \subseteq B_{C_t}(0) \tag{23}$$

*for all $t \in [0, \tau]$, where $C_t = (R + 2\mathscr{M}t)e^{3\mathscr{M}t}$.*

*Proof.* By definition of the Lagrangian solution (17),

$$\|X(t, x)\|_2 \leq \|x\|_2 + \int_0^t \|\mathcal{F}(X(s, x), \mu(s))\|_2 \, ds$$

$$\leq \|x\|_2 + \int_0^t \mathscr{M}(1 + \|X(s, x)\|_2 + M_1(\mu(s))) \, ds \tag{24}$$

Integrating both sides of (24) with respect to $\mu_0$ and noting that $X(t, \cdot)_{\#}\mu_0 = \mu_t$, we get,

$$M_1(\mu(t)) \leq M_1(\mu_0) + \mathscr{M} \int_0^t (1 + 2M_1(\mu(s))) \, ds$$

Combining this with (24) itself we get,

$$\|X(t,x)\|_2 + M_1(\mu(t)) \leq M_1(\mu_0) + \mathcal{M} \int_0^t (2 + \|X(s,x)\|_2 + 3M_1(\mu(s)))\, ds$$

This implies

$$\|X(t,x)\|_2 + M_1(\mu(t)) \leq (M_1(\mu_0) + 2\mathcal{M}t) + \int_0^t 3\mathcal{M}\left(\|X(s,x)\|_2 + M_1(\mu(s))\right)\, ds.$$

We note that the above equation is in the integral form of Gronwall's lemma ($u(t) \leq \alpha(t) + \int_a^t \beta(s)u(s)\, ds$) with $\alpha(t) = M_1(\mu_0) + 2\mathcal{M}t$ and $\beta(s) = 3\mathcal{M}$. Therefore, an application of Gronwall's lemma gives us,

$$\|X(t,x)\|_2 + M_1(\mu(t)) \leq (M_1(\mu_0) + 2\mathcal{M}t)e^{3\mathcal{M}t}$$
$$\leq (R + 2\mathcal{M}t)e^{3\mathcal{M}t}$$

Hence $\operatorname{supp} \mu^{\mathcal{F}} \subseteq B_{C_t}(0)$. $\qquad\square$

**Theorem 4.12.** *Suppose $\mu_0 \in \mathcal{P}_c(\Omega)$ is such that the support of $\mu_0$ is contained within $B_R(0) \subset \Omega$, for some $R > 0$. Additionally, let $K$ satisfy $K > (R + 2\mathcal{M}\tau)e^{3\mathcal{M}\tau}$. Furthermore, assume that*

$$\|\mathcal{T}_n(x,\mu)\|_2 \leq \mathcal{M}\left(1 + \|x\|_2 + M_1(\mu)\right), \tag{18}$$

*for all $x \in B_K(0)$ and all $\mu$ with support in $B_K(0)$. Then, there exists a unique Lagrangian solution to the approximate continuity equation (8) such that $\operatorname{supp} \mu^{\mathcal{T}}(t) \subseteq B_{C_t}(0)$ for all $t \in [0, \tau]$, where $C_t = (R + 2\mathcal{M}t)e^{3\mathcal{M}t}$.*

*Proof.* Let $\phi \in C_c^\infty(\mathbb{R}^{(n+1)d})$ be a compactly support smooth function such that $\phi(x) = 1$ for all $x \in B_{R+C_\tau}^{(n+1)}(0)$. We construct the function $\hat{T}[x,\mathbf{z}] = \phi(x,\mathbf{z})T[x,\mathbf{z}]$. Let $\hat{\mathcal{T}}_n$ be the expected transformer corresponding to $\hat{T}$. We check that $\hat{\mathcal{T}}_n$ satisfies Assumption 4.3*a)*. Towards, this end we compute

$$\|\hat{\mathcal{T}}_n(x,\mu) - \hat{\mathcal{T}}_n(y,\nu)\|_2 = \left\| \int_{\Omega^n} (\hat{T}([x;\mathbf{z}]))_1\, d\mu^{\otimes n}(\mathbf{z}) - \int_{\Omega^n} (\hat{T}([y;\hat{\mathbf{z}}]))_1\, d\nu(\hat{\mathbf{z}}) \right\|_2$$

Let $\gamma \in \mathcal{P}(\mathbb{R}^d \times \mathbb{R}^d)$ be the optimal plan with marginals $\mu$ and $\nu$ that solves the optimization problem (12). Then

$$\|\hat{\mathcal{T}}(x,\mu) - \hat{\mathcal{T}}(y,\nu)\|_2 = \left\| \int_{\Omega^n \times \Omega^n} (\hat{T}([x;\mathbf{z}]))_1\, d\gamma(\mathbf{z},\hat{\mathbf{z}}) - \int_{\Omega^n \times \Omega^n} (\hat{T}([y;\hat{\mathbf{z}}]))_1\, d\gamma(\mathbf{z},\hat{\mathbf{z}}) \right\|_2$$
$$= \left\| \int_{\Omega^n \times \Omega^n} (\hat{T}([x;\mathbf{z}]))_1 - (\hat{T}([y;\hat{\mathbf{z}}]))_1\, d\gamma(\mathbf{z},\hat{\mathbf{z}}) \right\|_2$$

The function $\hat{T}$ is globally Lipschitz for some Lipchitz constant $L$ as it is a product of a compactly supported smooth function $\phi$ and a locally Lipschitz function $T$. We can use this to conclude that

$$\|\hat{\mathcal{T}}(x,\mu) - \hat{\mathcal{T}}(y,\nu)\|_2 \leq L \int_{\Omega^n \times \Omega^n} \|x - y\|_2\, d\gamma(\mathbf{z},\hat{\mathbf{z}}) + L \sum_i^n \int_{\Omega^n \times \Omega^n} \|z_i - \hat{z}_i\|_2\, d\gamma(z_i,\hat{z}_i)$$
$$= L\|x - y\|_2 + L \sum_i^n \int_{\Omega^n \times \Omega^n} \|z_i - \hat{z}_i\|_2\, d\gamma(z_i,\hat{z}_i)$$
$$= L\|x - y\|_2 + Ln\mathcal{W}_1(\mu,\nu)$$

Therefore, $\hat{\mathcal{T}}$ satisfies Assumption 4.3*a)*. Then, Proposition 4.8 of Cavagnari et al. (2022) guarantees that there exists a unique Lagrangian solution $\mu^{\hat{\mathcal{T}}}(t)$ of (8) corresponding to $\hat{\mathcal{T}}$. Moreover, we know from Proposition B.1 that the support of the solution $\mu^{\hat{\mathcal{T}}}(t)$ lies in $B_{C_t}(0)$ for all $t \in [0, \tau]$. However, we note that $\hat{\mathcal{T}}(x,\mu) = \mathcal{T}(x,\mu)$ for all $x \in B_{C_t}(0)$ and all $\mu \in \mathcal{P}(\Omega)$ with support in $B_{C_t}(0)$. Therefore, $\mu^{\hat{\mathcal{T}}}(t) = \mu^{\mathcal{T}}(t)$, the unique Lagrangian solution of (8) for the expected transformer $\mathcal{T}$. $\qquad\square$

# C   Proof of Theorem 4.14

**Theorem 4.14** (Mean Field Dynamics Approximation Using Transformers). *Let $\varepsilon > 0$ be small enough and $n \geq 1$. Suppose that $\mathcal{F}$ satisfies Assumption 4.3a) for some $p$. Assume that the support of $\mu_0$ is contained within $B_R(0) \subset \Omega$, for some $R > 0$, and let $K \in \mathbb{R}$ be such that $K > (R + 2\mathcal{M}\tau)e^{3\mathcal{M}\tau}$. If the transformer $\mathcal{T}_n$ satisfies the condition*

$$\|\mathcal{T}_n(x, \mu) - \mathcal{F}(x, \mu)\|_* < \varepsilon.$$

*for all $z \in B_K(0)$ and $\mu \in \mathcal{P}(B_K(0))$. Then we have that*

$$\mathcal{W}_1(\mu^{\mathcal{F}}(t), \mu^{\mathcal{T}_n}(t)) \leq \varepsilon t \exp(2\mathcal{L}t) \tag{19}$$

*where $\mu^{\mathcal{F}}$ and $\mu^{\mathcal{T}_n}$ are the solutions to (2) and (8), respectively, and the estimate (19) is independent of $\mu_0$.*

*Proof.* Using the uniform norm approximation $\|\mathcal{T}_n(x, \mu) - \mathcal{F}(x, \mu)\|_* < \varepsilon$ and Assumption 4.3b), we can conclude that

$$\|\mathcal{T}_n(x, \mu)\|_2 \leq (\mathcal{M} + \varepsilon)(1 + \|x\|_2 + M_1(\mu))$$

for all $x \in \mathbb{R}^d$ and $\mu \in \mathcal{P}(B_K(0))$. Since $K > (R + 2\mathcal{M}\tau)e^{3\mathcal{M}\tau}$, for $\varepsilon > 0$ small enough, and $n$ large enough we can conclude that, $K > (R + 2(\mathcal{M} + \varepsilon)\tau)e^{3(\mathcal{M}+\varepsilon)\tau}$. From Theorem 4.12, there exists a Lagrangian solution for (8) such that

$$\operatorname{supp}\mu^{\mathcal{T}_n}(s) \subseteq B_K(0)$$

for all $s \in [0, \tau]$. Due to Assumption 4.3 and Proposition 4.8 of (Cavagnari et al., 2022), there exists a unique Lagrangian solution for (2). Once again, using Proposition B.1 we can conclude that

$$\operatorname{supp}\mu^{\mathcal{F}}(s) \subseteq B_K(0)$$

for all $s \in [0, \tau]$. Let $X, Y$ be the flow maps associated with with respect the vector fields $\mathcal{F}$ and $\mathcal{T}_n$, respectively.

From the definition of Lagrangian solutions we know that,

$$X(t, x) = x + \int_0^t \mathcal{F}(X(s, x), \mu^{\mathcal{F}}(s))ds$$

$$Y(t, x) = x + \int_0^t \mathcal{T}_n(Y(s, x), \mu^{\mathcal{T}_n}(s))ds$$

for all $t \in [0, \tau]$. From this we get

$$\|Y(t, x) - X(t, x)\|_2 = \left\| \int_0^t \mathcal{T}_n(Y(s, x), \mu^{\mathcal{T}_n}(s))ds - \int_0^t \mathcal{F}(X(s, x), \mu^{\mathcal{F}}(s))ds \right\|_2$$

$$\leq \left\| \int_0^t \mathcal{T}_n(Y(s, x), \mu^{\mathcal{T}_n}(s))ds - \int_0^t \mathcal{F}(Y(s, x), \mu^{\mathcal{F}}(s))ds \right.$$

$$\left. + \int_0^t \mathcal{F}(Y(s, x), \mu^{\mathcal{F}}(s))ds - \int_0^t \mathcal{F}(X(s, x), \mu^{\mathcal{F}}(s))ds \right\|_2$$

By triangle inequality of the $\|\cdot\|_2$ norm,

$$
\leq \left( \left\| \int_0^t \mathcal{T}_n(Y(s,x), \mu^{\mathcal{T}_n}(s))ds - \int_0^t \mathcal{F}(Y(s,x), \mu^{\mathcal{F}}(s))ds \right\|_2 \right.
$$

$$
\left. + \left\| \int_0^t \mathcal{F}(Y(s,x), \mu^{\mathcal{F}}(s))ds - \int_0^t \mathcal{F}(X(s,x), \mu^{\mathcal{F}}(s))ds \right\|_2 \right)
$$

$$
\leq \left\| \int_0^t \mathcal{T}_n(Y(s,x), \mu^{\mathcal{T}_n}(s))ds - \int_0^t \mathcal{F}(Y(s,x), \mu^{\mathcal{F}}(s))ds \right\|_2
$$

$$
+ \left\| \int_0^t \mathcal{F}(Y(s,x), \mu^{\mathcal{F}}(s))ds - \int_0^t \mathcal{F}(X(s,x), \mu^{\mathcal{F}}(s))ds \right\|_2
$$

$$
\leq \left\| \int_0^t \mathcal{T}_n(Y(s,x), \mu^{\mathcal{T}_n}(s))ds - \int_0^t \mathcal{F}(Y(s,x), \mu^{\mathcal{T}_n}(s))ds \right\|_2
$$

$$
+ \left\| \int_0^t \mathcal{F}(Y(s,x), \mu^{\mathcal{T}_n}(s))ds - \int_0^t \mathcal{F}(Y(s,x), \mu^{\mathcal{F}}(s))ds \right\|_2
$$

$$
+ \left\| \int_0^t \mathcal{F}(Y(s,x), \mu^{\mathcal{F}}(s))ds - \int_0^t \mathcal{F}(X(s,x), \mu^{\mathcal{F}}(s))ds \right\|_2
$$

Using the fact that $\|\cdot\|_2$ is convex, applying Jensen's inequality yields,

$$
\leq \int_0^t \left\| \mathcal{T}_n(Y(s,x), \mu^{\mathcal{T}_n}(s))ds - \int_0^t \mathcal{F}(Y(s,x), \mu^{\mathcal{T}_n}(s)) \right\|_2 ds
$$

$$
+ \int_0^t \left\| \mathcal{F}(Y(s,x), \mu^{\mathcal{T}_n}(s))ds - \int_0^t \mathcal{F}(Y(s,x), \mu^{\mathcal{F}}(s)) \right\|_2 ds
$$

$$
+ \int_0^t \left\| \mathcal{F}(Y(s,x), \mu^{\mathcal{F}}(s))ds - \int_0^t \mathcal{F}(X(s,x), \mu^{\mathcal{F}}(s)) \right\|_2 ds
$$

Using the fact that

$$
\operatorname{supp} \mu^{\mathcal{T}_n}(s) \subseteq B_K(0)
$$

we get,

$$
\|Y(t,x) - X(t,x)\|_2 \leq \int_0^t \varepsilon ds
$$

$$
+ \int_0^t \mathscr{L}\mathcal{W}_1(\mu^{\mathcal{F}}(s), \mu^{\mathcal{T}_n}(s))ds + \int_0^t \mathscr{L}\|Y(s,x) - X(s,x)\|_2 ds.
$$

Integrating with respect to $\mu_0$ and noting that $\mu_0$ is a probability measure,

$$
\mathcal{W}_1(\mu^{\mathcal{F}}(t), \mu^{\mathcal{T}_n}(t)) \leq \varepsilon t
$$

$$
+ \int_0^t \mathscr{L}\mathcal{W}_1(\mu^{\mathcal{F}}(s), \mu^{\mathcal{T}_n}(s))ds + \int_0^t \mathscr{L}\mathcal{W}_1(\mu^{\mathcal{F}}(s), \mu^{\mathcal{T}_n}(s))ds
$$

$$
\leq \varepsilon t + 2 \int_0^t \mathscr{L}\mathcal{W}_1(\mu^{\mathcal{F}}(s), \mu^{\mathcal{T}_n}(s))ds.
$$

Now, applying Gronwall's inequality, we get,

$$
\mathcal{W}_1(\mu^{\mathcal{F}}(t), \mu^{\mathcal{T}_n}(t)) \leq \varepsilon t \exp(2\mathscr{L}t)
$$

$\square$

# D  Transformer

**Definition D.1** (Multi-Headed Self-Attention). Let $X \in \mathbb{R}^{n \times d}$ be a matrix whose rows are $n$ data points in $\mathbb{R}^d$. Let $W_Q, W_K, W_V \in \mathbb{R}^{d \times d}$ be learnable weight matrices. Define the *query*, *key*, and *value* matrices by

$$Q = XW_Q, \quad K = XW_K, \quad V = XW_V.$$

Let $\mathrm{softmax}$ denote the softmax function applied row-wise to a matrix. The self-attention head function $\mathrm{AttHead} : \mathbb{R}^{n \times d} \to \mathbb{R}^{n \times d}$ is defined as

$$\mathrm{AttHead}(X) := \mathrm{softmax}\left(\frac{QK^\top}{\sqrt{d}}\right) V.$$

Let $h \in \mathbb{Z}_+$ be the number of attention heads. Let $\mathrm{AttHead}_1, \dots, \mathrm{AttHead}_h$ be attention heads with their own weight matrices, and let $W_0 \in \mathbb{R}^{hd \times d}$ be a learnable weight matrix. The multi-head self-attention layer $\mathrm{Att} : \mathbb{R}^{n \times d} \to \mathbb{R}^{n \times d}$ is defined as

$$\mathrm{Att}(X) := [\mathrm{AttHead}_1(X), \dots, \mathrm{AttHead}_h(X)] \, W_0,$$

where $[\cdot]$ denotes concatenation along the feature dimension.

**Definition D.2** (Transformer Network). A *transformer block* $\mathrm{Block} : \mathbb{R}^{n \times d} \to \mathbb{R}^{n \times d}$ is defined as

$$\mathrm{Block}(X) := X + \mathrm{FC}\left(X + \mathrm{Att}(X)\right),$$

where FC are feed-forward layers (position-wise fully connected layers), and $\mathrm{ReLU}$ is the rectified linear unit activation function. The addition operations represent residual connections. Let $L \in \mathbb{Z}_+$, and let $\mathrm{Block}_1, \dots, \mathrm{Block}_L$ be transformer blocks. A *transformer network* $T : \mathbb{R}^{n \times d} \to \mathbb{R}^{n \times k}$ is defined as a composition of transformer blocks followed by an output network:

$$T(X) := \mathrm{FC}_{\mathrm{out}}\left(\mathrm{Block}_L \circ \mathrm{Block}_{L-1} \circ \cdots \circ \mathrm{Block}_1(X)\right),$$

where $\mathrm{FC}_{\mathrm{out}} : \mathbb{R}^{n \times d} \to \mathbb{R}^{n \times k}$ is a fully connected neural network applied position-wise.

# E  Training

Here we provide the training details.

## E.1  Learning the Vector Field

**Hyperparameters** We consider depths in $\{3, 4, 5\}$, widths in $\{128, 256, 512\}$, and learning rates in $\{0.0002, 0.0001, 0.001\}$.

For the kernel method, we use polynomial basis of $\{1, x, \dots, x^d\}$ and sine and cosine basis of $\{\sin(x), \sin(2x), \dots, \sin(kx)\}$ and $\{\cos(x), \cos(2x), \dots, \cos(kx)\}$. We search over $d \in \{2, 3, 4\}$ and $k \in \{3, 4, 5\}$.

**Training Details** We used Adam with a cosine annealing decay rate for the step size. For the synthetics CS data, we used a batch size of 500 and trained the model for 1000 epochs. For the fish milling data, we used a batch size of 1, and trained the model for 10 epochs.

## E.2  Simulating dynamics

For the simulating dynamics experiment, we also train a transformer to learning the training dynamics of a two layer neural network. We fix the transformer to have a hidden dimension of 512 and 5 layers. We train the model for 250 epochs, using a learning rate 0.0002, batch size of 1000, using Adam with cosine annealing.

