# OpenReview forum: "Universal Approximation of Mean-Field Models via Transformers"
_ICML.cc/2025/Conference — ICML 2025 poster_

### Official Review · Reviewer_NEo9 · 2025-02-23

**Overall Recommendation:** 4

**Summary:**

The papers consider a mild variant of the transformer model, as part of a larger literature connecting transformers and maps to/from probability measures.  Their main result, from my perspective, is Theorem 4.14 which provides a sort of "small time" approximation guarantees that their version of the transformer model can efficiently approximate certain MF ODEs.  This is largely a consequence of their main technical result (Theorem 4.7).

I find that, the title perhaps mis-matches the claims, as when reading I was expecting to see approximation guarantees for MFGs or interacting particle systems.  Perhaps ODEs should be **clearly** placed in the paper's title and abstract, and not "models" which is far too overpromissing.

**Claims And Evidence:**

Rigorous and correct proofs.

**Essential References Not Discussed:**

The authors compare extensively to Furuya et al. (2024); however, the same author has since provided improved, i.e. fully quantitative results without passing to a "non-continuous limit" of attention in:
- "Is In-Context Universality Enough? MLPs are Also Universal In-Context." (2025).
Can you please compare to more contemporary result of the same author, and not old results?

Additionally, the authors should comment on the relationship to the probabilistic transformer of
- Kratsios A, Zamanlooy B, Liu T, Dokmanić I. "Universal Approximation Under Constraints is Possible with Transformers." International Conference on Learning Representations (Spotlight).
which is measure-valued and, which also yields a universal vector-valued map, when the expectation/barycenter map is applied at the output layer.

**Experimental Designs Or Analyses:**

Effectively NA

**Methods And Evaluation Criteria:**

Good

**Other Comments Or Suggestions:**

NA

**Other Strengths And Weaknesses:**

I don't understand the point of Assumption 4.3 in proving Theorem 4.14.  If $\Omega$ is compact then for any Borel probability measure $\mu \in \mathcal{P}(\Omega)$ and any $p>0$ we have
$$
\mathbb{E}_{X\sim \mu}[\|X\|^p] \le \operatorname{diam}(\Omega)^p,
$$
whence $\mu\in \cap_{p>0}\mathcal{P}_p(\Omega)$; so why the subscript.  Also, if we're in the metric setting, which we are, then not consider the weakest condition where $p=1$?

**Questions For Authors:**

- Can you shed some light on how to interpret (2) given that the second argument of $\mathcal{F}$ on measures, and there is a derivative of $\mu$ in time (for non-experts in MFGs).

- A key step in the proof of Theorem 4.7 in the quantization results of Fournier et al.  Can the authors provide LBs by combining the LBs in [1] with those in [2] into their analysis.

[1] Kloeckner, Benoit. "Approximation by finitely supported measures." ESAIM: Control, Optimisation and Calculus of Variations 18.2 (2012): 343-359.

[2] Ronald A DeVore, Ralph Howard, and Charles Micchelli. Optimal nonlinear approximation.
Manuscripta mathematica, 63(4):469–478, 1989


- Can the authors comment on the aforementioned transformer results and theirs?  Especially the more recent Furuya et al. (2025) result?

**Relation To Broader Scientific Literature:**

NA

**Theoretical Claims:**

Not always careful; e.g.
- In Assumption 4.3, what range are you allowing for p?  $p>0$, or I assume $p\ge 1$.  Can $p$ be infinite (e.g. compactly supported measures)?  The same issue is in the first line of the proof of Theorem 4.14.

Otherwise the proofs seem correct

---

> ### Author Rebuttal · Authors · 2025-03-31
>
> We thank the reviewer for the comments and feedback, and we thank the reviewer for finding that our results show that "transformer model can efficiently approximate certain MF ODEs." We hope that this response answers the reviewer's concerns.
>
> > Assumption 4.3
>
> Assumption 4.3 a) is an assumption on $\mathcal{F}$ map and not on the measures. If we restrict to compactly supported measures, arbitrary maps $\mathcal{F}$ may not satisfy the Lipschitz condition for any $p$. Hence, we assume that the map satisfies the Lipschitz condition for some *finite* p.
>
> Thus, for any $p$ for which we have the Lipschitz condition, Theorem 4.14 holds for that $p$.
>
> > Comparison with Kratsios and Furyaya  (KF) 2025 and Kratsios et al. (K) 2022
>
> We thank the reviewer for pointing us to these interesting papers. In the interest of fairness, *we would like to point out that KF 25 follow-up paper was posted to arxiv on 5th February after the ICML deadline on 30th January.*
>
> **Role of the Measure**
>
> 1. Our Work: The measure $\mu$ is an input argument representing the state distribution in a mean-field system, directly influencing the vector field $\mathcal{F}(z, \mu)$. This directly models physical or biological system interactions. A key strength is its applicability to general Borel probability measures $\mathcal{P}(\Omega)$ on a compact set $\Omega$.
>
> 2. K 22: The measure $\mathbb{P}$ is the output of the model $\hat{F}(x)$, explicitly designed to handle constraint satisfaction by ensuring the output distribution lies within the constraint set $K$.
>
> 3. KF 25: The measure $\mu$ is an input argument to the target function $f(\mu, x)$. However, it's restricted to the specific class of Permutation-Invariant Contexts $\mathcal{P}\_{C,N}(\mathcal{X})$ within a geometrically constrained domain $\mathcal{K}$, aimed at analyzing general in-context function approximation.
>
> **Map Definition and Transformer**
>
> 1. Our Work: Defines the map $\mu \mapsto \mathcal{T}_n(\cdot, \mu)$ (Measure to Vector Field) using the Expected Transformer $\mathcal{T}_n$, derived by taking the expectation of a standard, finite-sequence transformer $T$. This provides a practical link between standard architectures and measure-theoretic inputs.
>
> 2. K 22: Defines the map $x \mapsto \hat{F}(x)$ (Vector to Output Measure) using a modified transformer incorporating Probabilistic Attention, explicitly outputting a measure.
>
> 3. KF 25: Defines the map from finite vector spaces to finite vector spaces. However, the input and outputs are interpreted as measures on discrete sets.
>
> 4. Connection via Thm 4.7: If the map approximated in K \& F satisfies our assumptions, then our Theorem 4.7 could be applied to the transformer $\hat{\mathcal{T}}$ constructed in K \& F's Corollary 5.
>
> **Guarantees**
>
> 1. Our Work: Provides quantitative $L_\infty$ bounds on the vector field approximation error ($\|\mathcal{T}_n - \mathcal{F}\|$) that explicitly show convergence as the number of particles $n$ increases, linking the error to the quality ($\mathcal{E}$) of the underlying finite transformer. Furthermore, it connects this to the approximation of the system's dynamics ($\mathcal{W}_p$ bounds for $\mu(t)$) via Gronwall's lemma.
>
> 2. K 22: Guarantees exact support constraint satisfaction ($supp(\hat{F}(x)) \subseteq K$) combined with quantitative $W_1$ bounds on how well the output measure approximates the target.
>
> 3. KF 25: Provides quantitative probabilistic $W_1$ bounds on the output approximation error, dependent on the target function's modulus of continuity $\omega$ and the domain's geometry $q$, focusing on the network size needed for a given precision $\epsilon$.
>
> > Interpret (2)
>
> Mean field Equations of the kind in (2) are used for modeling the dynamics of a large system of interacting particles. Here, the derivative in time models the change in distribution (or measure) of the particles, and there is not necessarily any underlying variational aspect to their evolution. In this way, mean field games (MFGs) are distinct, where the evolution of the agents are dependent on the measure, through a loss function of a coupled game.
>
> > Lower Bounds
>
> The reviewer is correct that using this, we can lower bound the quantity in Line 535. However, this is only one of the three terms that we bound. Even if we manage to individually bound the terms, we used a variety of inequalities, such as Jensen's and triangle inequality, to get our decomposition.
>
> > Title
>
> Please note that the particle models we consider are formulated as ODEs (e.g. Eq. 6), while the mean-field models (e.g. Eqs. 2 and 7) are expressed as PDEs. In Theorem 4.14, we demonstrate that solutions of the continuity equation can be approximated by the *approximate continuity equation*, where the transformer replaces $\mathcal{F}$. In essence, our work shows that mean-field models derived from both ODEs and PDEs can be approximated, therefore the word *models* aligns nicely with the overall theme of our study.

---

> > ### Comment · Reviewer_NEo9 · 2025-04-06
> >
> > Dear authors, I have finished reading there rebutal and am satisfied with it. Thanks for the clear response :)

---

### Official Review · Reviewer_6g3z · 2025-02-26

**Overall Recommendation:** 4

**Summary:**

The authors study how transformes can be used to approximate mean-field models. The analysis is both theoretical and empirical.
Empirically, they test the transformers' on two different mean field models. Theoretically they provide bounds in therms of $L_\infty$ distance between the expected transformer and the mean field vector field.

**Update after rebuttal**

I am satisfied with the authors' answer and confirm my positive evaluation of the paper thus recommending its acceptance

**Claims And Evidence:**

yes

**Essential References Not Discussed:**

not that I am aware of

**Experimental Designs Or Analyses:**

yes. The experimental anaysis looks sound and valid to me

**Methods And Evaluation Criteria:**

yes

**Other Comments Or Suggestions:**

The authors are invited to provide a more extensive comparison with the work of Furuya 2024 in the introduction already as it seems to be very related to what they are doing.

The sentence "Additionally, Furuya et al." use a continuous version of transformes and attention and provide universal approximation of measure theoretic maps is too minimalistic and does not explain what is the substantial difference between their work and your work.

-In the introduction the symbols $P(\Omega)$ and $D(\Omega)$ are undefined

**Other Strengths And Weaknesses:**

I think that studying how transformers can be used to approximate mean field games has very important implications in machine learning.
The paper looks original to me and the question the authors try to answer is fundamental.

**Questions For Authors:**

The authors are invited to provide a more extensive comparison with the work of Furuya 2024 in the introduction already as it seems to be very related to what they are doing.

**Relation To Broader Scientific Literature:**

Yes, there is a dedicated paragraph in the introduction where the authors mention several works where mean field games are learned from the particle level dynamics. Moreover, in section 4 the authors make more specific comparison with a work in the literature in terms of theoretical resutls obtained.

**Theoretical Claims:**

I checked the proof of the main theorem (Theorem 4.7) in Appendix A and looks correct to me

---

> ### Author Rebuttal · Authors · 2025-03-31
>
> We thank the reviewer for the comments and feedback. We thank the reviewer for finding that the problem we study has "very important implications in machine learning" and that "the paper looks original to me and the question the authors try to answer is fundamental". We hope that this response answers the reviewer's concerns.
>
> > Comparison with Furaya et al 2024
>
> We acknowledge the reviewer's comment regarding the need for a more detailed comparison with the work of Furuya et al. (2024) in the introduction.
>
> Furuya et al. (2024) establish that deep transformers are universal approximators for general continuous 'in-context' mappings defined on probability measures. Their measure-theoretic approach defines transformers directly on the space of probability distributions, leveraging a continuous version of attention. A key result is that a single deep transformer architecture, with fixed embedding dimensions and a fixed number of heads, can approximate any such continuous mapping uniformly, even when dealing with an arbitrarily large or infinite number of input tokens represented by the measure.
>
> Our work, while also connecting transformers to measure theory, takes a different approach tailored to approximating the specific structure of mean-field dynamics in interacting particle systems. Instead of defining a continuous transformer directly, we utilize standard transformers designed for finite sequences and propose a novel lifting mechanism: the 'Expected Transformer' ($\mathcal{T}_n$). This construct maps the standard finite-particle transformer to the space of measures via an expectation over the particle distribution. Our primary goal is not general function approximation on measures, but rather to specifically approximate the vector field $\mathcal{F}$ governing mean-field dynamics and, consequently, the evolution of the system's distribution described by the associated continuity equation. **Hence Theorems 4.14 is a major contribution of our work.**
>
> The substantial difference lies in both the model definition and the nature of the approximation guarantee. While Furuya et al. provide an existence result for a single, deep, fixed-dimension transformer achieving a target precision $\epsilon$ for arbitrarily many tokens, our work provides quantitative approximation bounds for the Expected Transformer $\mathcal{T}_n$. These bounds explicitly characterize how the approximation error for the infinite-dimensional vector field $\mathcal{F}$ depends on two factors: (i) the approximation quality ($\mathcal{E}$) of the underlying *finite-dimensional* transformer on $n+1$ particles, and (ii) the number of particles $n$ used in the expectation, leveraging known convergence rates of empirical measures in Wasserstein distance. We further utilize these bounds to establish guarantees on approximating the *solution trajectories* of the mean-field continuity equation, linking the vector field approximation error to the error in the dynamics via stability results like Gronwall's inequality.
>
> Below are the specific substantial differences highlighted:
>
> **Approximation Target**
>
> 1. Furuya et al. Aim to approximate *general continuous in-context mappings* $\Lambda^*(\mu, x)$ defined on probability measures. Require target mapping $\Lambda^*$ to be *continuous* w.r.t. weak* topology (plus Lipschitz conditions on contexts for masked case). Their focus is broad representational power.
>
> 2. Our Work: Specifically targets the approximation of the *vector field* $\mathcal{F}(z, \mu)$ governing *mean-field dynamics* and the subsequent approximation of the *dynamical system's evolution* (solution to the continuity equation). Requires target vector field $\mathcal{F}$ to be *Lipschitz continuous* w.r.t. spatial and measure arguments (using Wasserstein distance).
>
> **Transformer Definition:**
>
> 1. Furuya et al.: Define transformers directly on the space of probability measures using a *measure-theoretic formulation* with continuous attention layers ($\Gamma_{\theta}(\mu, x)$)
>
> 2. Our Work: Uses standard transformers $T$ designed for *finite sequences* of length $n+1$. Introduces the "Expected Transformer" $\mathcal{T}_n(x, \mu)$ which *lifts* the finite transformer's output to the measure space via an expectation operation.
>
> **Handling Input Size (Number of Tokens/Particles):**
>
> 1. Furuya et al. Show a *single* transformer architecture (with fixed dimensions/heads) works uniformly for an *arbitrary* number of input tokens (even infinite) for a given precision $\epsilon$.
>
> 2. Our Work: The approximation quality of $\mathcal{T}_n$ explicitly *improves as n increases*, reflecting empirical measure convergence. Focus is on convergence behavior.
>
> > In the introduction the symbols are undefined
>
> We thank the reviewer for pointing this out, we have now added the definitions.

---

### Official Review · Reviewer_svzQ · 2025-03-13

**Overall Recommendation:** 3

**Summary:**

This paper shows, both empirically and with theoretical guarantees, that mean-field dynamics ("transport-type" dynamical systems over the space of probability measures, i.e., which take the form of a continuity equation $\partial_t \mu_t = -\nabla_z \cdot (\mu_t \mathcal{F}(z,\mu_t))$) can be approximated up to any finite time horizon using transformers, provided the vector field $\mathcal{F}$ is Lipschitz-continuous in the Wasserstein sense. This is achieved by:
- fixing a number $n$ of particles and considering the $n$-particle dynamics corresponding to the desired mean-field dynamics, $\frac{d}{dt} \mathbf{z}\_t = \mathcal{F}(z, \nu^n\_{\mathbf{z}\_t})$ where $\mathbf{z}\_t \in (R^d)^n$ and $\nu^n\_{\mathbf{z}\_t} = \frac1n \sum\_{i=1}^n \delta\_{\mathbf{z}\_{ti}}$
- choosing a transformer model $T_\theta: \Omega^{n+1} \to R^{(n+1) \times d}$, where $\Omega \subset R^d$ is the domain over which we consider probability measures ($\Omega = R^d$ in the experiments and $\Omega=$a compact set in the theory sections)
- learning a transformer $T = T_{\hat{\theta}}$ that approximates the mapping $\Omega^{n+1} \ni \mathbf{z} \mapsto \left[ \mathcal{F}(\mathbf{z}\_1, \nu^{n+1}\_{\mathbf{z}}, ..., \mathcal{F}(\mathbf{z}\_{n+1}, \nu^{n+1}\_{\mathbf{z}} \right]$
- considering the vector field, called expected transformer, $\mathcal{T}\_n: (x, \mu) \mapsto \mathbb E\_{\mathbf{z} \sim \mu^{\otimes n}} T([x, \mathbf{z}])$, and using the associated dynamics $\partial_t \mu\_t = -\nabla\_z \cdot (\mu\_t \mathcal{T}\_n(z,\mu_t))$ as an approximation of the desired mean-field dynamics.

On the experimental side, this methodology is validated on two toy examples (the Cucker-Smale and the Fish Milling models) in dimension 4, and on approximating the training dynamics of two-layer neural networks (in the mean-field parametrization).

On the theoretical side, the paper provides quantitative estimates of the approximation error for the proposed scheme. In particular the approximation error upper bounds vanish when $\Omega$ is compact, the time horizon is fixed, and $n$ goes to infinity. These theoretical guarantees rest upon previous works' results on the approximation power of transformers as sequence-to-sequence mappings.


## update after rebuttal

See discussion in the comments. I decided to keep my score of 3, though it's a "strong" 3, because there are still some natural questions raised by this paper that are not really addressed, particularly the significance of the in-expectation result; this being said this is arguably okay for a conference publication.

**Claims And Evidence:**

Yes the claims are supported by clear and convincing evidence.

**Essential References Not Discussed:**

I am not aware of any essential references that were not discussed.

**Experimental Designs Or Analyses:**

I have not checked the soundness of experimental designs.

**Methods And Evaluation Criteria:**

Yes the proposed methods and evaluation criteria make sense.

**Other Comments Or Suggestions:**

It seems to me that the approach taken in this work could be generalized to the case where the map $\mathcal{F}(z,\mu)$ is of the form $\mathcal{G}(z,\mu) + \nabla \log \mu(z)$ where $\mathcal{G}$ is Wasserstein-Lipschitz (in the sense of Assumption 4.3a). That is, it could be generalized to maps containing an isotropic diffusion term. This would connect to the use of transformers for score learning in the context of score-based diffusion models.

If the target dynamical system $\partial_t \mu_t =-\nabla_z \cdot (\mu_t \mathcal{F}(z, \mu_t)$ converges (or is stable), then we can hope for a uniform-in-time (resp. polynomial-in-time) approximation error. I wonder if the proposed methodology achieves this, and if this could be shown using this paper's tools. This remark is inspired by the favorable observed behavior for the simulation on the Cucker-Smale model in dimension 4.

**Other Strengths And Weaknesses:**

See my comment on the use of the expected transformer in "Relation to Broader Scientific Literature".

**Questions For Authors:**

- Please address the minor technical concerns listed in "Theoretical Claims".
- Do you know of any practical ML settings in which learning a mean-field dynamics from a finite number of observed trajectories is directly relevant?
- The approximation bounds given in the paper are for the dynamics using the expected transformer. In practice, how can the expected transformer mapping be computed? Can the permutation invariance of $T$ be exploited for fast computation?
- To complement the empirical comparison of the proposed method's performance compared to baselines in section 3.1: how do these methods compare in terms of computational cost?

**Relation To Broader Scientific Literature:**

The paper shows end-to-end guarantees for the problem of approximating mean-field dynamics using transformers. The possibility of using transformers for this purpose is not surprising given the universal approximation capabilities of transformers as sequence-to-sequence maps, but this paper gives a complete rigorous analysis. The error bounds obtained in this paper are relatively straightforward consequences of classical techniques in the context of mean-field dynamics, and are likely way too pessimistic, but it is still worthwhile to write those bounds down properly, which this paper does well.

Compared to related works, in particular Furuya et al. (2024), this paper takes an alternative and arguably simpler approach. Indeed, by considering the expected transformer (and using an easy triangle inequality argument, line 516), this paper's approach allows to apply previous finite-sequence-length approximation results directly "off the shelf". (I am not aware of other works taking this approach, but I am not familiar with the literature on transformers.)

**Theoretical Claims:**

I have checked Appendix A (proof of Thm 4.7) and Appendix C (proof of Thm 4.14). Both contain minor mistakes (or missing steps I wasn't able to fill) which may slightly impact the constants in the bounds:
- The last step of the proof of Thm 4.7 concludes to an upper bound which is different from the one in the theorem statement: $\mathcal{L} \frac{2}{n+1}$ instead of $C \mathcal{L} \mathrm{diam}(\Omega)^p n^{p/q-1}$.
- In the proof of Thm 4.14, I don't see how line 704 is obtained: if it is again by Young's inequality then there should be an extra $2^{p-1}$ factor a priori. On line 717, I don't see how the first term can be upper-bounded by $\int_0^t \varepsilon ds$, it seems it should be $(\int_0^t \varepsilon ds)^p$, and similarly for the two other terms. Why not just consider $\\|Y(t,x)-X(t,x)\\|_p$ and use triangle inequalities? Even then, the norms used in the inequalities stated in Assumption 4.3 don't seem to be sufficient to have dimension-independent bounds at this step of the proof. By the way, this proof assumes that in the definition of the distance $W_p$, the distance $d$ is the $p$-norm, which is not necessarily standard, so it may be useful to specify it. For simplicity it might be preferable to stick to using the $2$-norm on $R^d$ and considering $W_p$ distances defined as in Definition 4.2 with $d$ being the $2$-norm, and in the proof of Thm 4.14, bound $\\|Y(t,x)-X(t,x)\\|_2$.

The proof of Corollary 4.10 appears to be missing.

---

> ### Author Rebuttal · Authors · 2025-03-31
>
> We thank the reviewer for the comments and feedback. We thank the reviewer for finding that "the paper shows end-to-end guarantees" and that it is  "worthwhile to write those bounds down properly, which this paper does well," and for finding that the paper takes a novel, "arguably simpler approach" compared to prior work.
>
> We hope that this response answers the reviewer's concerns.
>
> > Last step of Thm 4.7 proof, upper bound different from theorem statement
>
> The reviewer is right, we fixed the upper bound. It now reads:
>
> $$\left\|\mathcal{T}_n - \mathcal{F} \right\|\_{*} \le \mathcal{E} +  \mathscr{L} \text{diam}(\Omega)^{p} \left(\frac{1}{n+1} + CG(n,p,q) \right) $$
>
> > Technical details for theorem 4.14
>
> The reviewer is correct on all counts.
>
> We were missing a factor of $2^{p-1}$ in line 704.
>
> The term should be $\int_0^t \varepsilon^p ds$. Moreover, we did miss a dimension-related term : $d^{\frac{p}{2}}$ that is due to the conversion between $\|\cdot\|_\infty$ and $\|\cdot\|_2$ norms. The final result that we obtain is:
>
> $$ \mathcal{W}^p\_p(\mu^{\mathcal{F}}(t),\mu^{\mathcal{T}\_n}(t))  \leq 2^{2p-1} d^{\frac{p}{2}} \varepsilon^p t\exp(\mathscr{L}^p 2^{2p-1} d^{\frac{p}{2}} t). $$
>
> We don't need Young's. We use $(a+b)^p \leq 2^{p-1}(a^p+b^p)$ with the triangle inequality to give us the needed result.
>
> We now use the classical definition using $\|\cdot\|_2$.
>
> > Corollary 4.10 proof missing
>
> We thank the reviewer for pointing this out. Here is the sketch: The proof follows from Theorem 4.3 of Alberti et al., which states -
>
> For each permutation equivariant function $f$ and $\epsilon > 0$, there exists a Transformer $T$ such that
>
> $$ \sup\_{X \in \mathcal{X}^n}\|f(X) - T(X)\|\_\infty < \epsilon $$
>
> Although the statement does not explicitly provide bounds on the sizes, these can be inferred from the construction. Specifically, the architecture comprises a two-layer network described by Hornik (1989), followed by a single attention layer, and then another two-layer feedforward network from Hornik (1989), resulting in constant depth. The result of Hornik et al. (1989) does not impose bounds on the widths of these networks. The single attention layer has a width of $1+2d+d'$, where $d'$ remains constant with respect to $d$.
>
> > Generalize to maps containing isotropic diffusion term
>
> We thank the reviewer for this insightful comment. This is something we are currently looking at. While approximation of score functions using transformers is a potential application, the score function is not Lipschitz in the measure, so one requires additional work since results of our paper don't immediately extend to this case.
>
> > Uniform-in-time (resp. polynomial-in-time) approximation error
>
> This a good point, however, this would require the notion of stability on Wasserstein spaces and robustness of stable mean-field systems to perturbations, so it would involve more work to extend ideas from approximation of classical ODE theory to approximation of mean-field ODEs.
>
> > Learning mean-field dynamics from a finite number of observed trajectories
>
> This work is relevant to generative modeling or sampling problems in which a noise distribution is transported to a target distribution.
>
> Another interesting application is training neural networks. We can train a smaller network, approximate the training dynamics using the Transformer, and then increase the width (i.e., increase the number of particles). Then, we could train the large-width model using the Transformer. Note that this doesn't require knowing the training data.
>
> > Approximation bounds for the dynamics using the expected transformer
>
> In practice, the expected transformer can be approximated quickly. For example, if $x$ is a data point and $\mu$ is the measure. Suppose we have $B$ collections $z^{(1)}, \ldots, z^{(B)}$ of particles, where each $z^{(i)}$ is state of $n$ particles. Then, we can append $x$ to each of the $B$ collections.
>
> For the transformer, we represent the input with a batch size $B$ and a sequence length of $n+1$. This allows the forward pass to be efficiently parallelized in any modern ML library. After obtaining the outputs, we compute their mean to approximate the expected transformer output. Instead of calculating the theoretical mean, we use the sample average over $B$ samples, concentrating around the true mean at a rate proportional to the transformer's variance divided by $B$. We expect that a reasonably sized $B$ will yield an accurate estimate.
>
> We did this for the Cucker Smale model. The results can be seen in Figure 4. Here, we used $B = 10000$. On a single GPU, this takes a few seconds.
>
> > Time Complexity
>
> This is a great question.
>
> For the fish milling dataset, on a single L4 GPU: Transformer: 6-10 minutes per model,
>
> TransformerConv: 5-10 minutes per model,
>
> Cyclindrical, FNN, and Kernel were all under 2 minutes for the largest model.
> The smaller models were on the order of seconds.

---

> > ### Comment · Reviewer_svzQ · 2025-04-04
> >
> > Re "Technical details for theorem 4.14":
> > - It seems to me that dimension-independent bounds can be obtained if the Euclidean norm is used instead of the $|.|\_1$ and $|.|\_\infty$ norms in the definition of the regularity assumption, Assumption 4.3 (please correct me otherwise). It might simplify the computations (and make comparison with related works easier) to stick with the Euclidean norm throughout (this is a minor technical point that is entirely up to you, as perhaps there are applications which I am not aware of for which the current form of Assumption 4.3 is more convenient).
> >
> > Re "Uniform-in-time (resp. polynomial-in-time) approximation error": Fair point. I would personally be curious to see in the future what guarantees can be obtained, especially since most dynamics one is typically interested in (including those in your numerical experiments actually) do exhibit some kind of stability or convergence.
> >
> > Re "Approximation bounds for the dynamics using the expected transformer":
> > - Do I understand correctly that the $B$ collections of particles, $z^{(1)}, ..., z^{(B)} \in (R^d)^n$, are also updated throughout the run? Using what vector field? My current understanding is that they would need to be updated using the $z^{(1)}, ..., z^{(B)} \in (R^d)^n$ themselves, so the concentration of the sample average becomes unclear, as we lose independence after one iteration.
> > - Is there any way to bound the transformer's variance theoretically? Empirically, does the variance appear to be uniformly bounded (across iterations, datapoints, etc.)?

---

> > > ### Author Response · Authors · 2025-04-06
> > >
> > > > Dimension-independent bounds can be obtained if the Euclidean norm is used
> > >
> > > The reviewer is right, and we thank them for their comments, these changes make our calculations more consistent, and improve the readability.
> > >
> > > We proved the equivalent versions of Theorems 4.7 and 4.14 when $\|\cdot\|\_\infty, \|\cdot\|\_p$ are replaced by the $\|\cdot\|\_2$ norm. For the new versions we use $\mathcal{W}\_1$ metric with the $\|\cdot\|\_2$ norm.
> > > The new assumptions would read:
> > >
> > > $$ \| \mathcal{F}(x,\mu) - \mathcal{F}(y,\nu) \|\_{2} \leq \mathscr{L} \left( \| x - y \|\_{2} + \mathcal{W}\_1(\mu, \nu) \right).$$
> > >
> > > and
> > >
> > > $$\| \mathcal{F}(x,\mu) \|\_{2} \leq \mathscr{M} \left( 1 + \| x \|\_{2} + M\_1(\mu) \right), $$
> > >
> > > The new norm now reads:
> > >
> > > $$\| \mathcal{H} \|\_{star} := \sup_{x \in \Omega} \sup\_{\mu \in \mathcal{P}(\Omega)} \| \mathcal{H}(x, \mu) \|\_{2}.$$
> > >
> > > The reviewer is right, by changing to $\|\cdot\|\_2$, the dimension-dependent term in the proof of Theorem 4.14 can be removed. The new bound in Thm 4.14 now reads:
> > >
> > > $$ \mathcal{W}\_1(\mu^{\mathcal{F}}(t),\mu^{\mathcal{T}_n}(t)) \leq  \varepsilon t\exp(2\mathscr{L} t) $$
> > >
> > > > Re "Approximation bounds for the dynamics using the expected transformer":
> > >
> > > The reviewer's understanding is correct, we would have to update using the Transformer. Hence, the reviewer is correct that the independence only holds for the first iteration. However, empirically, at least for small time horizons with $B=1$ (see Figures 1,2,3), the lack of independence doesn't seem to be an issue.
> > >
> > > However, making a proper comparison for the expected transformer beyond the first iteration is tricky, as we would need the ground truth continuous measure which is non-trivial to compute.
> > >
> > > For the first iteration, empirically, the variance seems to be 2 orders of magnitude smaller than the mean. For our Cucker Smale model, we observe a mean on the order of 0.1 and a max variance of 0.003.

---

### Official Review · Reviewer_vHZs · 2025-03-14

**Overall Recommendation:** 3

**Summary:**

This paper explores the application of transformers in modeling the mean-field dynamics of interacting particle systems.  The study empirically shows that transformers can effectively approximate diverse mean field models, such as the Cucker-Smale model and systems for training two-layer neural networks. It supports these empirical findings with mathematical theory, proving that the approximation error between the transformer-based and the true mean-field dynamics can be quantified and is dependent on the number of particles used in training. Finally, it establishes theoretical bounds on these errors, enhancing the understanding of transformer capabilities in complex system dynamics.

**Claims And Evidence:**

Yes.

**Essential References Not Discussed:**

I am not an expert in this area so I am not sure.

**Experimental Designs Or Analyses:**

I am not an expert in this area, but the benchmark experiments seem to be limited to models similar to Cucker-Smale, which I believe is not comprehensive enough.

**Methods And Evaluation Criteria:**

I am somewhat confused by Figure 1 and Table 1. Firstly, for the mean-field model, is it appropriate to solely use mean square error to assess performance? I was under the impression that it should also be evaluated by the distance between distributions.

**Other Comments Or Suggestions:**

See weakness.

**Other Strengths And Weaknesses:**

First of all, I feel really confused about the experiment results.

1. For table 1, I do not see any benefit of proposed method, regardless that I feel the distribution distance should also be reported. The proposed method seems to have small variance but the absolute mean is not competitive?

2. For figure 2, it makes me even more confused. Why the comparison is between SGD (optimizer) and Transformer (Model)? Am I missing anything here?

**Questions For Authors:**

See weakness.

**Relation To Broader Scientific Literature:**

N/A

**Theoretical Claims:**

No I did not. I am still confused with problem set up and the correctness of evaluation.

---

> ### Author Rebuttal · Authors · 2025-03-31
>
> We thank the reviewers for their questions and comments, which help improve the paper. We thank the reviewer for finding that our paper "supports these empirical findings with mathematical theory" and "establishes theoretical bounds on these errors, enhancing the understanding of transformer capabilities in complex system dynamics."
>
> We hope that our response can clear the reviewers' doubts.
>
> > I am somewhat confused by Figure 1 and Table 1. Firstly, for the mean-field model, is it appropriate to solely use mean square error to assess performance? ... it should also be evaluated by the distance between distributions.
>
> This is a great question: we have two points.
>
> 1. The MSE shown in Table 1 is about approximating the vector field $\mathcal{F}$. For a given distribution $\mu$, $\mathcal{F}$ is a map from $\Omega \times \mathcal{D}(\Omega) \subset \mathbb{R}^d \to \mathbb{R}^d$. Since the range of $\mathcal{F}$ is $\mathbb{R}^d$, using MSE is appropriate in this case. Alternatively, if we were minimizing the particle positions between the predicted model and data, it would have made sense to use 2-Wasserstein distance.
>
> 2. Additionally, if we were minimizing the particle positions between the predicted model and data, bounds on MSE imply bounds on the 2-Wasserstein distance but not vice versa. According to the definition,
>
> $ \mathcal{W}\_2(\mu,\nu)^2 = \inf\_{\gamma \in \Pi(\mu,\nu)} \int \|x-y\|^2 d\gamma(x,y).$
>
> The MSE bound restricts the integrand, thereby providing an upper bound on
> $ \mathcal{W}\_2(\mu, \nu)^2$.
> However, the reverse implication does not hold. For instance, consider two particles, $a\_1$ and $a\_2$, that both start at zero. With equal probability, either $a\_1$ moves to 1 and $a\_2$ to $-1$ or vice versa. At the level of distributions, these scenarios yield zero distance. However, the MSE between the two scenarios is 4.
>
> > For table 1, I do not see any benefit of proposed method, regardless that I feel the distribution distance should also be reported. The proposed method seems to have small variance but the absolute mean is not competitive?
>
> We apologize for the confusion; the $\times 10^{-k}$ factor multiplies both the mean and the variance. For example, for the Cucker Smale model, our method has a mean of $1.9 \times 10^{-6}$. The next best mean is the TransformerConv with $m = 20$ has a mean of $3.3 \times 10^{-6}$ which is nearly double. For the fish milling dataset, we have a mean of $2.2 \times 10^{-2}$ and the next best model is the TransformerConv with $m=3$ has a mean of $6.5 \times 10^{-2}$ which is nearly three times that of the Transformer. Thus, Table 1 shows that transformers have the best MSE error.
>
> > For figure 2, it makes me even more confused. Why the comparison is between SGD (optimizer) and Transformer (Model)? Am I missing anything here?
>
> We use the Transformer to approximate the ODE dynamics that govern the evolution of parameters in a two-layer neural network during training. That is, we are using the Transformer to train a distinct two layer neural network. This approach is analogous to the Cucker-Smale model, with the true dynamics defined by Equations (5) and (6). In this context, the true dynamics we aim to approximate are those induced by SGD. As Mei et al. (2019) demonstrated, the SGD dynamics can be expressed through a mean-field equation (Equation (7)), ensuring that our theory is directly applicable here.
>
> > I am not an expert in this area, but the benchmark experiments seem to be limited to models similar to Cucker-Smale, which I believe is not comprehensive enough.
>
> We consider three benchmark datasets. The first consists of simulated dynamics generated using the true Cucker-Smale equations. The second dataset features real-world observations of fish milling in a pond—actual fish behavior—which, although often conjectured to follow the Cucker-Smale model, is not guaranteed. The third dataset captures the dynamics of SGD for a two-layer neural network, representing a model that is notably distinct from the other two.

---

### Decision · Program_Chairs · 2025-05-01

**Decision:**

Accept (poster)

**Comment:**

This paper demonstrated that transformers can approximate the dynamics of mean-field models, systems in physics, biology, and modern machine learning. Through the concept of the expected transformer, the authors provide approximation guarantees with quantitative error bounds on the distance between the true mean-field dynamics and those obtained using the transformer. These results are complemented by careful empirical validation on both synthetic and real-world datasets.

Reviewers appreciated the novelty, depth, and clarity of the contribution. The authors have addressed reviewer concerns by correcting and improving the theoretical analysis, and providing precise comparisons with recent related work (e.g., Furuya et al., Kratsios et al.).

Overall, this is a high-quality paper with broad relevance across ML theory and applications. I recommend acceptance.